# Meta-Control: Automatic Model-based Control Synthesis for Heterogeneous Robot Skills

**Tianhao Wei**[*,1], **Liqian Ma**[*,1,2], **Rui Chen**[1], **Weiye Zhao**[1], **Changliu Liu**[1]
[*] Equal contribution, [1] Carnegie Mellon University, [2] Tsinghua University

meta-control-paper.github.io

**Abstract:** The requirements for real-world manipulation tasks are diverse and often conflicting; some tasks require precise motion while others require force compliance; some tasks require avoidance of certain regions, while others require convergence to certain states. Satisfying these varied requirements with a fixed state-action representation and control strategy is challenging, impeding the development of a universal robotic foundation model. In this work, we propose Meta-Control, the first LLM-enabled automatic control synthesis approach that creates customized state representations and control strategies tailored to specific tasks. Our core insight is that *a meta-control system can be built to automate the thought process that human experts use to design control systems.* Specifically, human experts heavily use a model-based, hierarchical (from abstract to concrete) thought model, then compose various dynamic models and controllers together to form a control system. Meta-Control mimics the thought model and harnesses LLM's extensive control knowledge with Socrates' "art of midwifery" to automate the thought process. Meta-Control stands out for its fully model-based nature, allowing rigorous analysis, generalizability, robustness, efficient parameter tuning, and reliable real-time execution.

**Keywords:** embodied agent, model-based control, LLM, manipulation

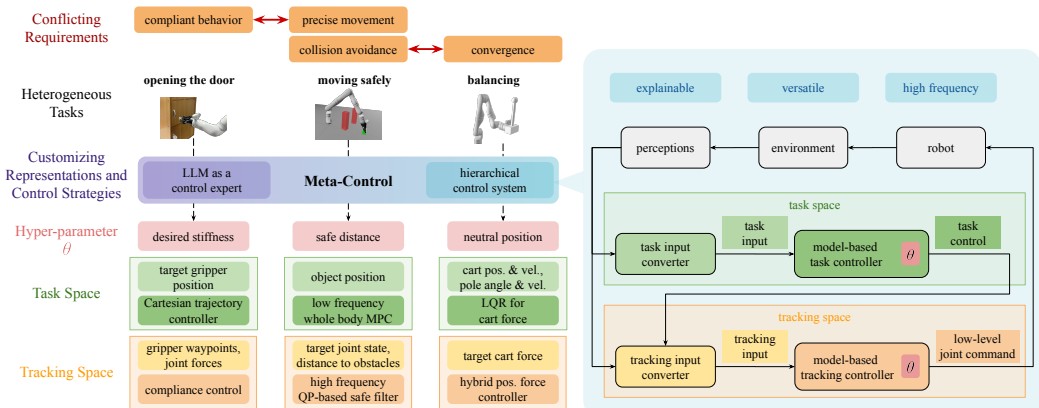

Figure 1: Opening the door, moving safely, and balancing are heterogeneous skills and have conflicting requirements that can be difficult to satisfy with a fixed control strategy and fixed state-action representation. Meta-Control addresses the challenge with a composable hierarchical control formulation and LLM, enabling automatic model-based control synthesis. The synthesized skill has customized state action representations, dynamic models, and controllers that perfectly align with the objectives and requirements.

8th Conference on Robot Learning (CoRL 2024), Munich, Germany.

# 1 Introduction

From control (for trajectory tracking) to learning (for trajectory generation), robotic systems have gained higher autonomy over the years. However, human designers still play an indispensable role in the creation and tuning of these control and learning algorithms in a case-by-case manner, which prevents the realization of general-purpose robots. For robots to better serve and assist human users in a variety of tasks, there is a pressing need for higher autonomy that allows robots to autonomously synthesize skills $S$ based on language instructions $\ell$ to any given task, without manual tuning. Some recent attempts have been made towards this goal using LLM. Such as automatically selecting a skill $S_i$ from a predefined skill library $\{S_1, S_2, \cdots\}$ [1, 2, 3], leveraging generic parameterized policies $S_\theta$, where $\theta := LLM(\ell)$ [4, 5, 6], and reinforcement learning (RL) with LLM generated rewards: $S := RL(LLM(\ell))$ [7, 8, 9]. However, the journey towards general-purpose robots, capable of adapting to arbitrary tasks without manual tuning, remains incomplete.

A major challenge for general-purpose robots is that real-world tasks have inherently diverse and often conflicting requirements that are difficult to satisfy with a fixed control strategy. For example, as illustrated in fig. 1, moving safely requires precise movements for collision avoidance, necessitating safe position control. Conversely, position control is unsuitable for opening the door due to the difficulty of planning a trajectory that perfectly aligns with the door's swing path, making force compliance control more favorable. Similarly, balancing demands controllers that ensure the pole's convergence (a position-attracting goal), which opposes collision avoidance (a position-avoiding goal). Although visuomotor methods can potentially address heterogeneous tasks, they often lack explainability and reliability. Predefined action primitives may cover different task types but require manual construction and have limited applicability and scalability. These challenges restrict existing methods from generalizing to various open-world manipulation tasks with varying constraints.

To advance towards a general-purpose robot, we propose Meta-Control, a novel framework $F$ that automatically synthesizes model-based robot skills from instructions for diverse open-world tasks: $S := F(\ell)$. Our core insight is that *higher level autonomy can be achieved by creating a meta-control system to automate the thought process that human experts use to design control systems, with the help of LLMs*. More specifically, human experts design different representations and control strategies for different tasks following a model-based, hierarchical (e.g., from abstract to concrete) thought model, and then compose various dynamic models and controllers into an integrated control system. By mimicking the thought model and the thought process, the autonomous system, Meta-Control, can gain greater coverage in robotics problems with good performance.

Based on these insights, we formulate robot skill design as a hierarchical control system synthesis problem and leverage LLM to customize representations and control strategies. The hierarchy involves a high-level task space and a low-level tracking space. The task space is an abstracted, intuitive space for accomplishing the task (e.g., Cartesian space or gripper pose space) while the tracking space usually represents the robot state space where low-level constraints can be specified and task-level commands are followed. Meta-Control first let the LLM determine the task and tracking spaces, then design the corresponding dynamic models and controllers utilizing predefined but abstract templates. The abstract model/controller template grounds the behavior of the synthesized system while maintaining high flexibility. We design a set of guidance and checklist of error in the prompt to help the LLM avoid potential errors and produce the correct code, just as how Socrate helps people draw their own conclusions.

Meta-Control offers several benefits: 1. It enables the synthesis of challenging heterogeneous robotic skills for unseen tasks, allowing each task to be accomplished with the most suitable representation and control strategy tailored to task-specific requirements. 2. Unlike previous work that primarily utilizes spatial priors from LLMs (e.g., object localization, affordance), Meta-Control leverages the internalized control knowledge of LLMs which encompasses decades of modeling and optimization efforts for various tasks and skills. 3. The synthesized control system is fully model-based which brings robustness, generalizability, efficient parameter tuning, and formal guarantees (e.g., safety and stability), leading to reliable and trustworthy execution.

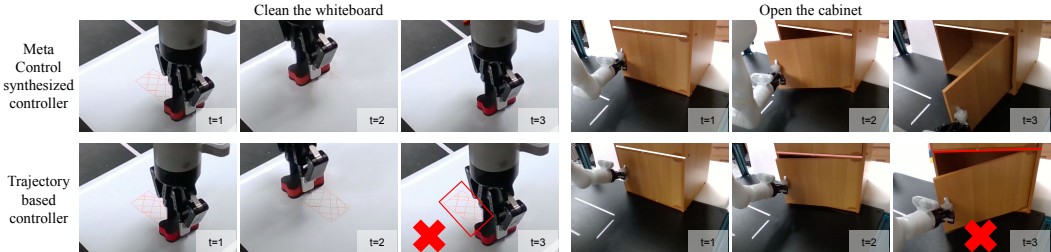

Figure 2: Comparison between Meta-Control and a trajectory planning-based method on a real robot for wiping the board and opening the cabinet. The trajectory-based method fails to erase the mark because it neglects force requirements. Opening the cabinet with a trajectory-based controller leads to cabinet displacement because the planned trajectory does not precisely align with the door's swing path, which may damage the door if the cabinet is fixed. In contrast, Meta-Control addressed these challenges with properly customized control systems.

## 2   Related work

We review four closely related approaches to achieve higher autonomy: skill libraries, optimization, end-to-end methods, and model-based control synthesis. Meta-Control integrates aspects of these approaches, offering a flexible, robust, and explainable solution for diverse real-world tasks.

**Skill libraries with LLM** methods utilize predefined libraries of motion primitives, enabling diverse control strategies through high-level APIs. LLMs can dynamically combine these libraries for task execution [1, 10, 11, 12, 13, 14, 15, 16, 17, 18, 19]. Skill libraries are often constructed using behavior cloning, reinforcement learning, or bootstrapping [5, 20, 21, 22, 23, 24, 25, 26, 27, 28, 29]. However, these motion primitives are manually constructed and task-specific. In contrast, Meta-Control is capable of synthesizing new heterogeneous skills on the fly.

**Optimization-based methods** use a fixed optimization framework such as MPC or RL with mutating hyper-parameters, including objectives [30, 31, 32, 33, 7, 8], constraints [34, 4, 35], or dynamic models [36, 37, 38, 39, 40, 41]. These hyper-parameters can be inferred by LLMs [4, 33, 7, 8] or learned from data [36, 30, 31]. Hierarchical optimization often use LLM inferred high-level commands for an instruction-following controller [42, 29]. A key limitation is that the chosen framework constrains the method's capability. An MPC generating end-effector actions cannot produce force-compliant control in joint space, and RL-based methods are unsuitable for online skill synthesis. Meta-Control, however, dynamically generates flexible policies.

**End-to-end models** directly map perceptions and instructions to robot actions, using VLMs [6, 43, 44, 45, 46, 47, 48, 49, 50, 51, 52] or energy functions and affordance maps [53, 54, 55, 56]. Diffusion models have also been used to learn from demonstration [57, 58, 20], enabling multimodal action distribution. End-to-end methods often output Cartesian actions, requiring additional motion planning for joint-level movement, and they are data-hungry. Furthermore, these methods lack robustness and explainability. Meta-Control is data-efficient, integrating LLM with model-based control strategies to ensure flexibility, robustness, and explainability.

**Model-based control synthesis** designs explainable systems with rigorous guarantees [59, 60, 61, 62, 63, 64, 65]. Hierarchical formulations simplify control synthesis for complex systems [62, 66, 67, 68, 69, 70, 71]. However, model-based synthesis is often task-specific and lacks generalizability. Meta-Control overcomes this limitation with the help of LLM.

## 3   Method

In this work, we focus on synthesizing robot skills, defined as unit actions (e.g., grab the eraser, erase the marks) from robot tasks instructed via free-form language $\mathcal{L}$ (e.g., clean the whiteboard) [4]. We assume that the decomposition from task to skills: $\mathcal{L} \rightarrow \ell_1, \ell_2, \ldots, \ell_n$ is given by a task-level

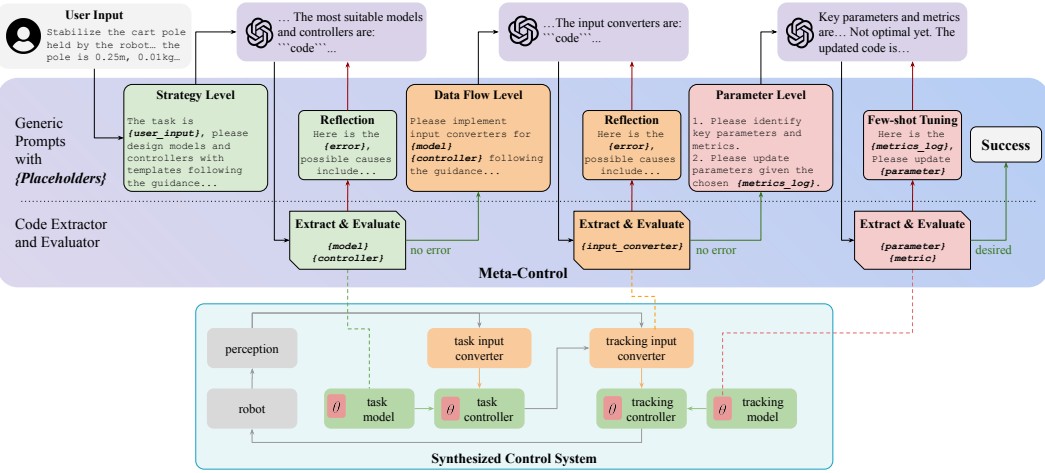

Figure 3: Overview of Meta-Control: The user only needs to provide a skill description. Meta-Control then leverages the control knowledge of LLMs to synthesize skills through a three-level pipeline: strategy level, data flow level, and parameter level. For each level, we have designed a generic prompt with placeholders, which are dynamically replaced with user input or code extracted from the LLM response during runtime, utilizing a code extractor to make the prompt task-specific. The extracted code is also used to construct the control system. At each level, if the LLM-generated code results in an error, a reflection phase is initiated. We have embedded design principles and checklists of common errors within the design and reflection prompts to assist the LLM in producing correct code. The generic prompt design with placeholders allows Meta-Control to generalize to unseen tasks without modification.

planner, which can be LLM-based or search-based [72]. Our focus is to synthesize a control system to perform a skill described by $\ell$.

Designing control systems from scratch for open-world skills is very difficult even for humans because the design space is infinite. Mimicking the thought model of control experts, we propose a composable hierarchical control formulation and introduce the Meta-Control pipeline.

### 3.1 Meta-Control Formulation: Composable Hierarchical Control

**Hierarchical Control** The bilevel hierarchical control formulation can represent various skills through a high-level controller in an intuitive task space and a low-level controller that tracks the high-level control in the robot state space. Formally, we define

$$\textit{State space:} \quad \dot{x} = f(x, u) \qquad \textit{Measurement space:} \quad y = g(x, u) \qquad \textit{Task space:} \quad \dot{z} = h(z, v)$$

where $x$ is the system state, $u$ is the state space control input, $y$ is the output or measurement of the system, $z$ is the state of the task space, and $v$ is the task space control input. The task space contains intuitive and high-level states, such as the gripper poses for robot arms and the center-of-mass for quadrupeds. $x$ and $z$ may not be directly observable, but $y$ has to be measurable or extractable from perception. $f, h$ are dynamic models of the corresponding system. We assume $g$, the perception algorithms, is given in this work, but $g$ could also be designed. The task space can sometimes be omitted when it coincides with the state space, such as when a robot arm skill directly specifies joint goals. We denote the task space controller by $\pi_v(y)$, and the tracking space controller by $\pi_u(y, v)$.

Our goal is to use LLM to 1) infer the objectives and constraints of the task; 2) design the proper task state $z$; 3) design the dynamic models $h(z, v)$ and $f(x, u)$; 4) design the task and tracking controller $\pi_v(y)$, $\pi_u(y, v)$ to achieve the desired performance.

**Composable Design** Designing $z$, $h$, $f$, $\pi_v$, and $\pi_u$ directly is still very challenging due to infinite possible spaces and dynamic models and the need for an accurate and deep understanding of the robotic system. Similar to humans, LLMs excel in making intuitive decisions, but are relatively poor at extensive reasoning, large-scale design, and implementation [73]. Furthermore, LLM gen-

erated code cannot guarantee constraint satisfaction. Therefore, we introduce model and controller templates, which enable composable design and guaranteed constraint satisfaction.

Templates are predefined object classes that need to be instantiated with concrete arguments. For example, we offer a dynamic model template called `LinearModel` which can be instantiated by passing in four matrices $A, B, C, D$ describing a linear system, and a controller template `LQRController` requires matrices $Q, R$ and vectors $x_0, u_0$. Templates differ from motion primitives in that they are abstract and generic. The usage of templates grounds the behavior of the system, greatly increasing explanability while maintaining flexibility.

## 3.2 Meta-Control Pipeline: Generic Prompts with Multi-Level Synthesis

As illustrated in fig. 3, we designed a 3-level pipeline for Meta-Control (Meta-Control) to generate hierarchical control systems using generic prompts that adapt to unseen tasks without modification. The user provides a task description, and Meta-Control interacts with an LLM in a Socratic manner to synthesize the control system. A detailed query and response can be found in appendix C.

**Art of Midwifery** LLMs often struggle to write fully functional code without errors. To guide it, we implemented a method similar to Socratic questioning, where instead of directly giving solutions, we prompt the LLM to "give birth" to correct code through structured guidance. Meta-Control interacts with the LLM through three dialogues, each representing a level of the pipeline, and after each, it generates a summary that informs the next step. These dialogues follow a structured template, with task-specific placeholders, filled in by user input or code extracted from the LLM's responses using a code extractor. This process ensures the pipeline can adapt to unseen tasks, simply by providing task descriptions. Each dialogue follows specific design principles, such as defining control spaces based on available measurements, designing the task controller with objectives in mind, and aligning the tracking controller with task requirements. If the generated code encounters errors, a reflection phase, depicted in fig. 3., is triggered. This phase uses a checklist of common LLM errors and related error messages to correct them. For example, Drake's [74] unusual rotation-translation order is a common source of errors. A detailed breakdown of prompt design techniques is provided in appendix B.

**Strategy Level: Design via Template** Meta-Control first queries the LLM to 1) identify the appropriate task space, objectives, and constraints; 2) selects the models and controllers from the template library, and 3) instantiate them with the correct parameters based on the task's requirements. The process follows a logical sequence: defining the task controller, selecting the task model, and then designing the tracking model and controller. The LLM's responses follow a strict format, allowing us to extract the generated code and evaluate it.

**Data Flow Level: Composition via Semantic Understanding** emplate inputs and outputs often differ in format and semantic meaning, so the LLM must align these interfaces to ensure unimpeded data flow. Meta-Control queries the LLM to implement two objects: 1) a task input converter that composes the task controller's input from available measurements $y$; 2) a tracking input converter that composes the tracking controller's input from $y$ and the task control $v$. F In the balance cart-pole example, the task input converter extracts the 4D input for the `LQRController` from 20+ available measurements. It then passes the task control (a scalar force) to the tracking input converter, which prepares the inputs for the `HybridPositionForceController`. The tracking input converter has to pad the task control and combine it with available measurements to form a 12D input. A checklist of common errors helps correct interface mismatches, such as dimensionality or data type issues.

**Parameter Level: Alignment via Few-Shot Optimization** The designed system may degenerate when deployed because the inferred parameters may not align the reality. To address this, Meta-Control enables few-shot optimization of system parameters by querying the LLM on 1) key parameters affecting behavior, 2) performance metrics, and 3) strategies for adjusting parameters based on past trajectories. For instance, in the cart-pole task, the LLM identifies the LQR controller's $Q$ and $R$ parameters as crucial, using the cart's position and pole's angle as metrics. The process is efficient because LLMs have internalized parameter-performance relationships in common dynamic

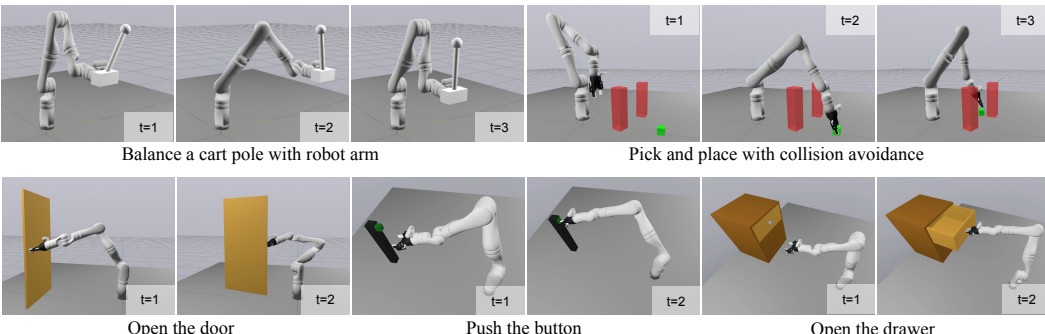

Figure 4: Five manipulation tasks that have inherently different challenges and requirements. For example, the balance task requires an accurate and high-frequency feedback controller. The safe pick and place task requires guaranteeing collision avoidance for the whole robot arm. The open door task requires properly handling articulated objects; and the executed trajectory has to perfectly match the swing path.

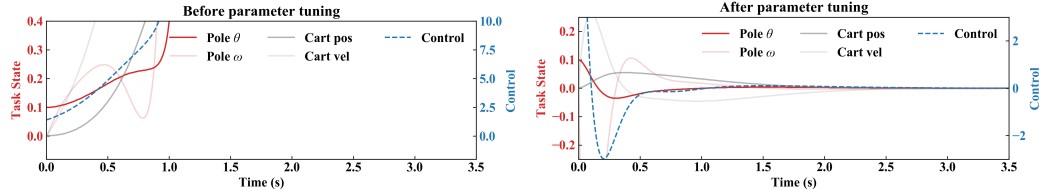

Figure 5: Meta-Control can automatically identify hyper-parameters that require tuning and tune them to accomplish challenging tasks. The figure shows the trajectory of the arm-held cart-pole system before and after tuning the synthesized controller. The hyper-parameters $Q = \text{diag}(10, 1, 100, 1)$, $R = 0.01$ are chosen and tuned by the LLM with only 2 rounds of trial-and-error.

systems. A list of possible root causes (e.g., initialization errors or suboptimal parameters) of the degenerated performance helps the LLM diagnose problems more effectively.

## 4 Experiment

The experiment is designed to manifest the following features. We first show that Meta-Control enables synthesizing skills for heterogeneous requirements by exploiting the control knowledge of LLM. Then we did an ablation study to show that all three levels of the Meta-Control pipeline improve the success rate; Finally, we reveal the benefit of the model-based design, including robustness, generalizability, and formal guarantees.

Most of the experiments are conducted on four representative tasks. These tasks cover constraints that appear frequently in the real world, such as constraints on position, velocity, force, compliance, convergence, and safety. We implement our pipeline with Drake[74], a framework designed for model-based control. For the hardware experiment, we used a Kinova Gen3 robot arm. The language model is GPT 4.0 with a default temperature 1.0.

### 4.1 Meta control enables synthesizing challenging skills

As shown in fig. 2 and fig. 4, Meta-Control successfully synthesizes controllers for various challenging tasks with inherently different requirements, both in simulation and in the real world. Figure 5 shows that Meta-Control can identify core metrics and critical parameters that affect performance, then efficiently and effectively tune the parameters based on the metric through trial and error. A detailed description of task challenges and synthesized control systems can be found in appendix A.1. The complete skill synthesis process can be found in appendix C.

**Meta-Control exploits dynamics priors** In contrast to previous work that focuses more on spatial relationship priors [1, 4, 5], Meta-Control exploits the LLM's internalized knowledge of dynamics, enabling rigorous synthesis of high-performance controllers. In the balance cart pole

| Method | | API | API + Hierarchy | API + Template | Meta-Control |
|---|---|---|---|---|---|
| | design | 30% | 90% | 60% | 100% |
| Balance | implementation | 0% | 30% | 20% | 90% |
| | execution | 0% | 0% | 0% | 70% |
| | design | 40% | 50% | 60% | 100% |
| Open door | implementation | 10% | 20% | 10% | 100% |
| | execution | 0% | 0% | 0% | 80% |
| | design | 0% | 0% | 40% | 90% |
| Safe Pick&place | implementation | 0% | 0% | 0% | 90% |
| | execution | 0% | 0% | 0% | 90% |

Table 1: The ablation study shows that hierarchical formulation (Hierarchy) and templates-based synthesis (Templates) both improves the *success rate*, compared to directly synthesis with only the low-level API. API + Template can be viewed as a generalized motion-primitive method.

task, LLM designs the task space dynamics $h(z, v)$ with a linear approximation in the form of $\dot{z} = Az + Bv$ around the upright position of the pole, where $A$ and $B$ are *correctly* given by the LLM: $A = [[0, 1, 0, 0]; [0, 0, \frac{m_{pole}g}{m_{cart}}, 0]; [0, 0, 0, 1]; [0, 0, \frac{g(m_{cart}+m_{pole})}{l_{pole}m_{cart}}, 0]]$, $B = [0; \frac{1}{m_{cart}}; 0; -\frac{1}{l_{pole}m_{cart}}]$, where $m$ denotes mass, $l$ denotes length, $g$ is the acceleration of gravity. Detailed description of the synthesized control system can be found at appendix A.1.

## 4.2 Ablation study of the Meta Control pipeline

To demonstrate the necessity of the hierarchy formulation and simplification using templates, we show the success rate of control system synthesis with different ways to query the LLM. Meta-Control use the query: "output is 7-DoF torque, input is ..." (API) + "consider a task space and a tracking space..." (Hierarchy) + "use available templates" (Template). We test the success rate in 3 stages: design (via templates), composition (interface alignment), and execution (after few-shot optimization). We say that a design is successful if the LLM-designed control system has the potential to perform the skill judged by an expert. We say that the implementation is successful if the system can run without errors. We say that the execution is successful if the control system finishes the skill as desired. We repeat each task 10 times to compute the success rate. Randomness is caused by the LLM. As shown in table 1, we can see that the success rates of all steps in the baseline are lower than Meta-Control. Although LLM gives reasonable architectures to finish the task in the Balance and Open Door task, the LLM fails to provide correct code and parameters to realize the control system due to the complexity of the system and the huge design space. With all the modules, we achieve the highest success rate for all tasks. The Safe Pick&Place task is especially difficult for the baseline because the baseline methods, even though prompted to avoid collision, were unable to successfully design a controller that can avoid collision continuously.

## 4.3 Meta-Control brings benefits of model-based design

The model-based nature of the synthesized controller brings a variety of benefits, such as robustness, explainability, generalizability, and rigorous analysis.

**Generalization to different attributes/states** Meta-Control synthesized control systems can easily generalize to scenarios of different attributes/states due to the model-based nature. Given a successfully synthesized control system, we test different attributes/states and calculate the success rate. The range of change and the results are shown in table 2. Examples are shown in fig. 6. Meta-Control achieved a 100% success rate for all scenarios.

**Rigorous Formal Analysis** The model-based design allows rigorous formal analysis for a variety of properties, making autonomous certification by LLM possible. For example, we can guarantee the convergence for the cart-pole stask by solving the Riccati equation, and ensure collision avoidance for the pick-and-place task by certifying a safety index. Details can be found in appendix A.3

| Balance | | Open Door | | Safe Pick&Place | |
| --- | --- | --- | --- | --- | --- |
| Pole Mass 0.01~0.5 kg | 10/10 | Handle Height 0.3~0.75 m | 10/10 | Obstacle Position 0.01~0.3 m (y-axis) | 10/10 |
| Cart Mass 0.05~0.5 kg | 10/10 | Handle Radius 0.3~0.7 m | 10/10 | Obstacle Size 0.1~0.45 m (height) | 10/10 |
| Initial Angle -0.5~0.5 rad | 10/10 | Door Mass 1~30 kg | 10/10 | Place Position -0.3~0.3 m (x-axis) | 10/10 |

Table 2: The synthesized controllers easily generalize to scenarios with different object states/attributes. The left column lists the range of the parameters for each scenario. The right column indicates the success count of 10 trials for each set of parameters.

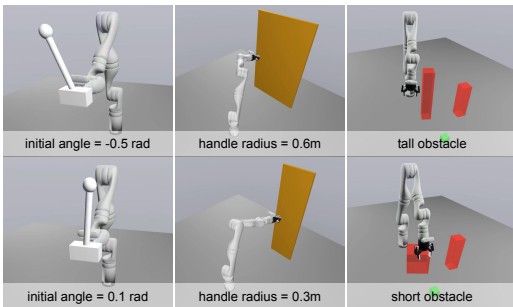

Figure 6: Meta-Control synthesized control systems are robust to attribute/state changes because of the model-based design.

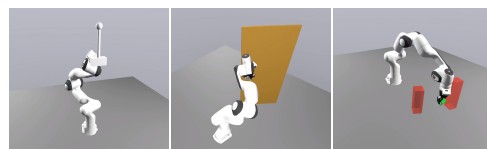

Figure 7: Meta-Control generalizes to different embodiments because the synthesis is fully model-based. Furthermore, a controller synthesized on Kinova can transfer to Franka Pranda simply by replacing the robot dynamic model thanks to the model-based nature.

**Transfer to real robot and different embodiments** The control system for opening door is synthesized in simulation and is executed both in simulation and in the real world. As shown in fig. 2 and fig. 4, the behavior is consistent and no sim-to-real gap is observed because the synthesized controller is model-based, closed-loop, highly explainable, and math-certified. The controller can also generalize to different embodiments with the same low-level API (e.g. 7 DoF joint torque) given the model of the new embodiments as shown in fig. 7.

### 4.4 Failure analysis

We analyze the failure cases in table 1 and summarize their reasons. 1. *Mathematical error:* Although LLM can give a mathematical description of the approximate dynamic model for an unseen system, it can make mistakes in math. For example, for the dynamic model synthesized for the cart-pole system, it can miss a term in the $A$ matrix, or mess up signs (use $+$ when $-$ is desired). 2. *Failure to follow instructions:* We require the LLM to provide a structured response so that a program can extract the code and plug it into the robotic system. However, sometimes LLM fails to follow the instructions, leading to responses in the wrong format. 3. *Incorrect reasoning:* In the open-door task, the LLM infers the target location of the door knob. However, the LLM may infer a wrong target given the environmental information. Although these are still challenging for LLMs, we believe that they can be overcome with the rapid development of LLMs in the near future.

## 5 Limitation and Discussion

In this work, we present Meta-Control, a novel framework for automatic model-based control system synthesis using LLMs, designed for heterogeneous robotic tasks. Simulations and real-world tests demonstrate its potential to enhance robotic autonomy. However, Meta-Control has limitations, such as reliance on accurate system state estimation and predefined models, which limit adaptability to new tasks. The LLM can also make suboptimal decisions, like selecting PID over LQR, complicating parameter tuning. Future improvements include better reflection across the pipeline, stronger reasoning capabilities, faster synthesis, and the integration of learning-based templates and automatic perception selection.

## 6 Acknowledgement

This work was supported by the National Science Foundation under Grant No. 2144489.

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

## A   Experiment details

### A.1   Task descriptions

*Open the door*    Opening a door is a challenge for robots because a door has a fixed swing path that must be followed exactly. As shown in fig. 2, position control can easily lead to door damage or failure of action. Therefore, it is preferable to open a door with a compliant controller. With multiple rounds of experiments, we found that Meta-Control synthesized control system usually involves a `CartesianStiffnessController` acting as the task controller or the tracking controller. Although the trajectory may not be perfectly aligned with the swing path, with the stiffness controller, the robot can still open the door because it complies with the force given by the door.

*Wipe the board*    Wiping a board requires a certain amount of force to be applied on the board while moving the eraser, which involves two different objectives: position tracking and force tracking. As shown in fig. 2, the synthesized controller successfully removes the marks, while control frameworks

| Task | Challenge | Meta-Control designed system |
|------|-----------|------------------------------|
| Open the door | The robot trajectory must *perfectly align* with the door's swing path. position control can easily lead to damage or failure. | CartesianTrajectoryController + CartesianStiffnessController, allowing imperfect trajectory planning and tracking with compliant behavior to avoid damage or failure. |
| Balance the cart pole | The pole is *non-actuated*. The system is sensitive, requiring high-frequency feedback and convergence guarantee. | LQRController + HybridPositionForceController. The LQR controller gives the force to be applied on the cart, and the hybrid position/force controller tracks the desired force on the y-axis while maintaining a neutral pose on the x-axis and the z-axis. |
| Collision-free pick and place | *whole-body* collision free in *continuous* time during the whole task. | KinematicTrajectoryMPC + SafeController, allowing discrete-time planning and continuous-time whole-body collision-free tracking. |
| Wipe the whiteboard | Two different objectives: tracking position and maintaining force | CartesianTrajectoryController + HybridPositionForceController, allowing position tracking while maintaining a desired force on the whiteboard. |

Table 3: Experiment tasks, challenges and Meta-Control synthesized controllers..

that only consider spatial relationships are not suitable for this task because of the lack of force constraints. In most trials, Meta-Control chooses a `CartesianInterpolationController` as task controller, and a `PoseForceController` as tracking controller. The Cartesian interpolation controller plans the trajectory of the eraser, while the hybrid position/force controller tracks the trajectory while maintaining a desired force on the board to erase.

*Balance the cart pole* Cart pole is a classic control task that has been extensively studied. Attempts were made to synthesize a simple PID controller with LLM to balance a pole with predefined APIs where the cart can be controlled directly [1]. However, in this experiment, we use a robot arm to hold the cart and ask LLM to balance it by controlling the robot arm. This is a significantly more challenging task because only low-level APIs of the robot arm are given, and the pole is attached to the cart with a *non-actuated free joint*. The LLM has to understand the relationship from the arm to the cart, and from the cart to the pole. In most cases, our method chooses an `LQRController` as the task controller and the `PoseForceController` as the tracking controller. The LQR controller gives the force to be applied on the cart along the pole joint direction (y-axis) to balance the pole, and the hybrid position/force controller tracks the desired force on the y-axis while maintaining a neutral pose on the x-axis and the z-axis. Profile of the pole's angle is shown in fig. 5, which shows that the synthesized controller efficiently balanced the pole.

The synthesized control system is described below:

$$y = [\text{Pole}_\theta, \text{Pole}_\omega, \text{Cart}_y, \text{Cart}_{\dot{y}}] \qquad x = \text{Joint states}, \text{EE}_{\text{force}}^{\text{target}}, \text{EE}_{\text{pose}}^{\text{target}}$$
$$v = \text{End-effector (EE) force on } y-\text{axis} \qquad u = \text{Joint torques}$$
$$h(z,v) = \text{Linearized Cart Pole dynamics} \qquad f(x,u) = \text{Kinova dynamics model}$$
$$\pi_v(y) = \text{LQR controller} \qquad \pi_u(x,y,v) = \text{Pose Force Controller}$$

where

$$h(z,v) = \begin{pmatrix} 0 & 1 & 0 & 0 \\ 0 & 0 & \frac{m_{\text{pole}}g}{m_{\text{cart}}} & 0 \\ 0 & 0 & 0 & 1 \\ 0 & 0 & \frac{g(m_{\text{cart}}+m_{\text{pole}})}{l_{\text{pole}}m_{\text{cart}}} & 0 \end{pmatrix} z + \begin{pmatrix} 0 \\ \frac{1}{m_{\text{cart}}} \\ 0 \\ -\frac{1}{l_{\text{pole}}m_{\text{cart}}} \end{pmatrix} v.$$

*Collision-free pick and place* Pick and place is a very common skill in daily life. In this task, we require the robot arm to reach a goal position while maintaining *whole-body* collision-free. The goal position can be either the location of the object or the target location. In most

cases, the LLM chooses a `KinematicTrajectoryModelPredictiveController` as the task controller to generate collision-free way-points for reaching the goal, and a `SafeController` as the tracking controller to guarantee collision-free in continuous time. In most cases, the LLM chooses a `KinematicTrajectoryModelPredictiveController` as the task controller to generate collision-free way-points for reaching the goal, and a `SafeController` as the tracking controller to guarantee collision-free in continuous time.

## A.2 Joint torques during openning a door

As shown in fig. 8, openning a door with position control can lead to very large joint torques, leading to failure or dangerous behaviors. But with a compliant controller, the joint torques are much smaller and the door is opened successfully without damage.

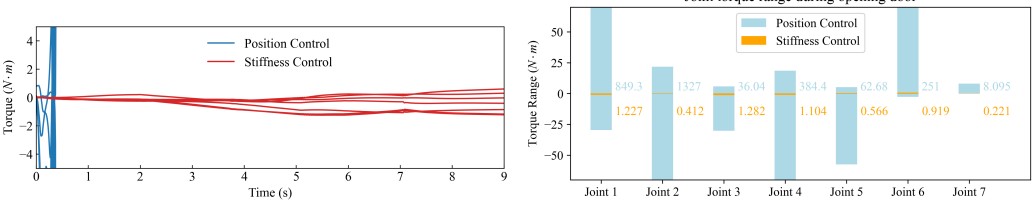

Figure 8: Joint torque range during opening door for the baseline (position control) and the Meta-Control synthesized controller (stiffness control). The baseline has a huge torque because the planned trajectory is inaccurate, which leads to damage.

## A.3 Formal Analysis of Properties

*Convergence and Stability:* For the balance cart-pole task, the convergence can be guaranteed by solving the Riccati equation for the LQR controller. The closed-loop system matrix $A - BK$ has the following four eigenvalues: $-412.29, -9.925, -1.502 + 1.175j, -1.502 - 1.175j$. All of them have negative real parts, which means that the system is guaranteed to converge. More rigorous analysis can be conducted by taking the linearization error into account.

*Constraint satisfiability and Forward Invariance:* In the pick-and-place task, an MPC task controller is tracked by a safe controller. The safe controller is realized with a safety index (also known as the barrier function), which guarantees collision avoidance with mathematical proofs [75, 62]: it ensures the system state always satisfies $\min\{d_{\min} - d(x), 100 \cdot (0.02^2 - d(x)^2) - 10 \cdot \dot{d}(x)\} < 0$, where $d_{\min}$ is the allowable minimum distance between the robot and the obstacle, $d(x)$ and $\dot{d}(x)$ are the relative distance and relative velocity from the robot to the obstacle, respectively.

## A.4 Runtime and parameter optimization rounds

In our experiments, it usually takes GPT-4o 1-3 minutes to synthesize a skill, depending on the capacity of the servery.

# B Prompt Design

We use a variety of techniques to improve the quality of the full dialogue. First, we introduce some techniques in the overall design, followed by the techniques used in each step of the three-level pipeline.

## B.1 Overall Design Techniques

1. Divide skill synthesis into three dialogues, corresponding to the three-level pipeline. This approach avoids overly long contexts, enables step-by-step reasoning, and allows reflections to focus on the current step.

2. At the end of each dialogue, Meta-Control asks the LLM to generate a summary to warm-start the next dialogue, including the decisions made in the current step and all necessary information for the next step.

3. Use a generic prompt with placeholder design to make the prompt zero-shot generalizable while remaining task-specific. Each placeholder is uniquely identified by a `<step_name>`. The placeholders are dynamically replaced by the LLM generated code.

4. Provide guidance and examples to help the LLM generate code following a specific format. The code should start with a `<step_name>`, e.g., `<task_model>`, followed by a Python code block encompassed by triple backticks (' ' ').

5. Use `<step_name>` to insert LLM-generated code into the control system. Our code extractor uses regular expressions to search for `<step_name>`, extract code from the LLM output, and then insert the code into the control system based on `<step_name>`.

6. Use a closed-loop design procedure. The LLM-generated code is executed by our program, which receives feedback, such as error messages or measurements during execution. Meta-Control then asks the LLM to improve the code based on the feedback.

7. Use code replacement instead of code completion to progressively construct the control system. Debugging and improvement are difficult with an incomplete control system code that is only executable after completing it with LLM-generated code, as feedback for intermediate steps would be unavailable. Therefore, we initially provide a blank but executable control system code, which outputs all zero joint torques. At each step of the synthesis, part of the code is replaced based on the identifier. We designed the control system to be always executable during the design process as long as the generated code is correct.

### B.1.1 Dialogue 1: Design via Template Prompt

1. Use a role-play prompt (e.g., a professor in control and a proficient programmer) to invoke control knowledge and coding skills.

2. Provide some optional principles to assist the design, such as "design the task controller on simplified dynamics" and "use the task controller to accomplish the primary objective while using the tracking controller to satisfy constraints or secondary objectives."

3. Use model templates and controller templates to ground the design space. The templates greatly reduce the design space and anchor the behavior of the synthesized control system, making it explainable and provable.

4. Provide all available measurements and their samples. The sample measurements give the LLM a precise idea of what inputs are available and their format, helping it to decide on the appropriate task controller and tracking controller.

5. Provide an example of the code that the LLM should return, including the `<step_name>` and how to initialize the chosen models and controllers.

6. Provide a checklist of requirements that the LLM should satisfy to reduce errors, such as "choices have been made for all steps" and "`<step_name>` must be included before each code block."

7. Use reflections. Meta-Control extracts and evaluates the code from the LLM response, then informs the LLM if the initialization of the chosen models and controllers causes any errors, allowing the LLM to decide if any fixes or improvements are needed. This process is repeated until there are no errors or the maximum iteration is reached.

8. Ask the LLM to generate a summary of the design to warm-start the composition dialogue. The summary includes the chosen models and controllers, a detailed description of the inputs for the controllers, the meaning of each dimension, and any other necessary information.

### B.1.2 Dialogue 2: Composition via Semantic Understanding

1. Use a role-play prompt as in the previous dialogue.

2. Use the design summary to warm-start the dialogue, providing the task, the chosen controllers and models, their input and output ports, and how they should be combined into a control system to complete the task.

3. Provide examples of how to access the values of the input and output ports of a model or controller.

4. Provide an example of the input converter, demonstrating how to connect the input and pad zeros when necessary. The input converter is a set of callback functions that assign values to the input based on the output of other modules.

5. Provide tips and potential errors to avoid when implementing the input converter. For example, "Ensure the dimensions of the output match the port. The output value may only be partially available from the input port," and "For pose, velocity, and force vectors, rotation always comes first."

6. Use partial updates. Because input converters involve many functions, if an error is detected during reflection, Meta-Control only updates the function that the LLM identifies as causing the error, instead of updating all functions, which may cause more errors if the code is not exactly the same.

7. Ask the LLM to generate a summary of the data flow to warm-start the parameter tuning, including parameters of the controller, parameters of the model, input converter structures, and potential performance metrics.

### B.1.3 Dialogue 3: Parameter Optimization

1. Use a role-play prompt as in the previous dialogues.

2. Provide a list of code blocks that the LLM can update to optimize the parameters, covering models, controllers, and input converters.

3. Warm-start the dialogue with the data flow summary, providing a detailed implementation of the control system, a description, and samples of the module inputs.

4. Provide a guide for parameter optimization: identify key parameters, identify performance metrics from task state and task control, and then determine if the time series of the metrics is desired.

5. Provide a checklist of errors to identify the cause of degraded performance, including definition, initialization, implementation, and suboptimal parameters.

6. Iteratively optimize the parameters with reflections. Every time the LLM updates the code or parameters, Meta-Control evaluates the new system, provides a time series of the chosen metrics, and allows the LLM to decide if the performance is desired. This process is repeated until the performance is satisfactory.

## C   Full conversation

The full conversation on skill synthesis is attached below. Some long numerical arrays are omitted for the sake of clarity. The controller templates, the dynamical model templates, and the input port samples are attached after the conversation.

## C.1 Balance

## C.1.1 Design via Templates

---

## Meta-Control
The GPT is a professor in robot control and a proficient programmer of Python and PyDrake. A robot task
↪ can be accomplished by sequentially executing several skills. The GPT will help the user write code to
↪ compose one of the skills after reading the requirements. The GPT will be instructed step by step. A
↪ skill is accomplished hierarchically by a task controller and a tracking controller. The task
↪ controller is responsible for generating high-level and abstract control, and the tracking controller
↪ is responsible for tracking the high-level control and satisfying constraints.

Here are some optional principles to design the task controller and the tracking controller:
1. Design the task controller for the object of interest or end-effector, and design the tracking
↪ controller for the robot.
2. Design the task controller on simplified or approximated dynamics, such as Cartesian space or an
↪ approximated linear model, and design the tracking controller on the full robot dynamics.
3. If the skill involves multiple objectives, design the task controller to take care of the primary
↪ objective and design the tracking controller to take care of the rest of the objectives and
↪ constraints. The task controller and the tracking controller can both output joint torque.
4. Task control is often the position, velocity, acceleration or force in the Cartesian space. But in some
↪ cases, the task control is best to be joint torque.
5. The final output of the tracking controller must be joint torques.

The GPT will proceed the following steps: task_controller, tracking_controller, task_model,
↪ tracking_model. In the GPT's response, the most important part will be <step_name> followed by a code
↪ block enclosed by ```python```. Each code block should correspond to a single step. This explicitly
↪ indicates the code is written for a specific step. The GPT can feel free to make some plans to help it
↪ think before writing the code blocks. There is no need to explain the code after the code block. The
↪ GPT should not import any module. The GPT should not make up arguments, variables, or anything else
↪ that is not mentioned.

## User
Please help me to compose a skill to stabilize a cart pole. The cart is attached to the end effector of
↪ the robot arm. The pole rotates freely around the x-axis and is unactuated. Please stabilize the pole
↪ around the upright position (when the pole angle, denoted by PolePin, is 0) by exerting force on the
↪ cart along the y-axis. A secondary objective is to keep the end-effector pose as close as possible to
↪ [1.57078469e+00, 0.0, 1.57079455e+00, 5.96598901e-01, 1.34625984e-03, 4.33589236e-01] ([roll, pitch,
↪ yaw, x, y, z]). The cart is 0.1 kg, and the pole is 0.25 m, 0.01 kg.

## Assistant
Certainly! I'm good at writing controllers for robots with PyDrake. I will follow your instructions
↪ carefully and think carefully.

## Meta-Control
Let's first take a look of all available measurements, dynamic models, and controllers.

Here are all available inputs to the skill:
{skill_inputs_sample}

Here are all available dynamic model setup functions in yaml format:
{dynamic_model_brief_str}

Here are all available controllers in yaml format:
{controller_brief_str}

=====================================================

Based on the available measurements, models, and controllers, please think step by step to choose the
↪ models and controllers:
1. Decide the subject of the task controller, is it an object, the end effector, or something else.
2. Decide the desired task control to be applied on the subject, is it Cartesian position, Cartesian
↪ force, or something else.
3. Decide the task space model that can model the dynamics of the subject with the task control. When the
↪ analytical form of the dynamics model is needed, draw upon your extensive knowledge of control theory
↪ and system modeling. Think step by step to make sure the model is correct according to the task
↪ description.
4. Decide the task controller that can apply the desired task control and realize the goal of the skill.
↪ Make sure the controller can realize the goal with the given available measurements. There are no
↪ additional inputs or functions provided to the controller.
5. Decide the tracking space model.
6. Decide the tracking controller to track the output of the task controller and to satisfy possible
↪ constraints.

Detail about how can the chosen models and controllers realize the skill goal. Be specific.

Please return the choice with the following format.

<task_model>

```python
task_model = setup_model(arg1, arg2)
```

<tracking_model>
```python
tracking_model = setup_model(arm_type, object_info)
```

<task_controller>
```python
task_controller = ControllerClass(arg1, arg2)
```

<tracking_controller>
```python
tracking_controller = ControllerClass(arg1)
```

Make sure your response complies with the following requirements:
* <step_name> must be included before each code block so the code block can be recognized. step_name must
  ↪ be enclosed in angle brackets <>.
* Choices have been made for all steps. A correct solution exists with the given information.
* Pass the arguments (args) when instantiating the controller and the model. The arguments needed are
  ↪ stated in the yaml file (args). **Do not** include items in "input_ports" (such as
  ↪ "controller_parameters") as arguments when instantiating the controllers. They are not arguments for
  ↪ instantiating the controller.
* The arguments should be defined before use, except `arm_type` and `object_info`, which can be used
  ↪ directly without definition.
* Do not use any placeholder or assume any variable is defined. All information and available variables
  ↪ needed are provided. If your choice of model and controller requires additional information, rethink
  ↪ it.
* The models and controllers are instantiated correctly as stated in the materials.
* The chosen model matches the required model of the controllers as stated in the materials.

## Assistant
Considering the requirements and the available models and controllers, let's proceed step by step to
↪ accomplish the task of stabilizing the cart pole attached to the end effector of the robot arm and
↪ achieving the secondary objective of keeping the end-effector's pose as close as possible to the
↪ specified position.

1. **Subject of the Task Controller:** Since the primary goal is to stabilize the pole around the upright
   ↪ position by exerting force along the y-axis to the cart, the task subject is "the cart and the pole
   ↪ system". This suggests we require a control method directly influencing the cart to alter the pole's
   ↪ position, making it a system-level control problem involving both the cart and the pole.

2. **Desired Task Control:** The control needed is a force along the y-axis on the cart, which indirectly
   ↪ influences the pole's angle to achieve stabilization.

3. **Task Space Model:** We need a model that accurately represents the dynamics of the cart-pole system
   ↪ for the application of a controlling force. Given the problem's nature, a linear model approximation
   ↪ around the upright position of the pole can be suitable. Therefore, I will choose the
   ↪ `setup_linear_model` because it can approximate the dynamics of the cart-pole system in an affine
   ↪ manner suitable for designing a controller to apply the required force.

4. **Task Controller:** To apply the desired task control (force), an appropriate choice is an
   ↪ `LQRController`. LQR (Linear-Quadratic Regulator) is well-suited for stabilizing the system around the
   ↪ equilibrium point (the upright position), given it requires a model of the system dynamics, which
   ↪ aligns well with our choice of a linear model.

5. **Tracking Space Model:** For the secondary goal, the model needs to incorporate the arm dynamics along
   ↪ with the constraints such as the end-effector's pose. The `setup_arm_with_objects_model` would be a
   ↪ suitable choice because it returns a model considering the robot arm and static objects, allowing
   ↪ control over the arm to maintain the end-effector's position while keeping the primary task of pole
   ↪ stabilization.

6. **Tracking Controller:** Considering we must respect both a desirable end-effector pose and exert
   ↪ control forces to stabilize the pole, a `PoseForceController` is selected for the tracking controller.
   ↪ This controller is capable of handling specific forces and poses, making it perfect for managing the
   ↪ secondary objective of maintaining the end-effector's pose.

Let's proceed with specifying the models and controllers as per our choice:

<task_model>
```python
x0 = np.zeros((4,1))  # Initializing the state for the cart-pole system
u0 = np.zeros((1,1))  # Initial control input (force)
m_cart = 0.1  # mass of the cart
m_pole = 0.01  # mass of the pole
l_pole = 0.25  # length of the pole
g = 9.81  # gravitational acceleration
```

```python
# A and B matrices are placeholders and must be defined based on the specific dynamics of the cart-pole
↪   system
A = np.array([[0, 1, 0, 0],
              [0, 0, m_pole * g / m_cart, 0],
              [0, 0, 0, 1],
              [0, 0, (m_cart + m_pole) * g / (l_pole * m_cart), 0]])
B = np.array([[0], [1/m_cart], [0], [-1/(l_pole*m_cart)]])
C = np.eye(4)
D = np.zeros((4, 1))
task_model = setup_linear_model(x0, u0, A, B, C, D)
```

<tracking_model>
```python
tracking_model = setup_arm_with_objects_model(arm_type, object_info)
```

<task_controller>
```python
Q = np.diag([1, 1, 10, 10])  # Emphasizing the importance of the pole's position and velocity
R = np.array([[0.1]])  # Control effort cost
task_controller = LQRController(task_model, Q, R, x0, u0)
```

<tracking_controller>
```python
force_control_axis_mask = np.array([False, False, False, False, True, False])  # Force control along the
↪   y-axis
tracking_controller = PoseForceController(tracking_model, force_control_axis_mask)
```

This sequence of models and controllers fulfills the skill's requirements by utilizing a systematic and
↪   hierarchical control structure.

## Meta-Control
Here is the output of running the skill: <empty>

Human expert supplementary instructions: <empty>

Do you think it is working as expected? (Empty output means the system is running without errors)
If it is, please answer with one word "yes".
If it is not, please identify which step(s) went wrong and explain how to fix it by including which
↪   step(s) to modify (<step_name>) and the updated code (enclosed by ```python```). We have proceeded
↪   with the following steps:
- task_model
- tracking_model
- task_controller
- tracking_controller

Please do not make up a step.

If the system is not working as expected, you must modify at least one step. You can modify multiple
↪   steps. Remember to add "<step_name>" before the corresponding code block. A correct solution exists
↪   with the given information. Try your best.

Here is a sample response when it is not working as expected:

I think this error occurred because [for some reason].

<tracking_controller>

updated_code:
```python
tracking_controller = NewTrackingController(arg1, arg2)
```

## Assistant
yes

## Meta-Control
Please summarize the choice of the task controller and the tracking controller. How would you expect the
↪   input output ports of the controllers to be connected to perform the skill? This response will be used
↪   as prompts to implement the port connector, so please include all necessary information, including:

* The detailed description of the input ports for the task controller and the tracking controller,
↪   including the meaning of each dimension (be careful about the order), and how to get them from the
↪   available measurements or outputs.
* A description of any constant parameter you have used that may affect the performance of the control
↪   system.
* Everything else you think is necessary to implement the controller by only reading the summary.

To implement the port connector for performing the skill of stabilizing a cart pole attached to the end
↪  effector of a robot arm, we selected two controllers: an `LQRController` for the task controller and a
↪  `PoseForceController` for the tracking controller. Here is the summary of how these controllers are
↪  expected to interact and how their inputs and outputs should be connected:

### Task Controller (LQRController)
- **Input Ports:**
  - `state`: It represents the current state of the cart-pole esystem. For the `LQRController`, this state
  ↪  vector is expected to have dimensions `[x, x_dot, theta, theta_dot]` where `x` and `x_dot` represent
  ↪  the position and velocity of the cart along the y-axis (as force is applied along the y-axis), and
  ↪  `theta` and `theta_dot` represent the pole's angle (around the x-axis) and angular velocity,
  ↪  respectively. This state can be constructed from the `measured_body_poses`,
  ↪  `measured_body_velocities` for the 'Cart', and 'PolePin' measurements from `measured_joint_states`.

- **Output Ports:**
  - `control`: This port outputs the computed control input based on the LQR control law. For our
  ↪  application, this control input is a single scalar value representing the force to be applied along
  ↪  the y-axis to the cart.

### Tracking Controller (PoseForceController)
- **Input Ports:**
  - `ee_pose_target`: This input requires the desired end-effector pose in the format `[roll, pitch, yaw,
  ↪  x, y, z]`. Since our primary goal is to keep the end-effector's pose as close as possible to a
  ↪  specified target while applying force to stabilize the pole, this target pose is set as a constant
  ↪  parameter `[1.57078469, 0.0, 1.57079455, 0.596598905, 0.00134625984, 0.483592036]`. Note that the
  ↪  force along the y-axis, which acts on the cart to stabilize the pole, is managed by this
  ↪  controller's `ee_force_target`.
  - `ee_force_target`: This input specifies the desired end-effector force in the format `[x-rotation,
  ↪  y-rotation, z-rotation, x-translation, y-translation, z-translation]`. The output from the task
  ↪  controller (LQRController) indicating the force to be applied along the y-axis to the cart will be
  ↪  fed into this port. All other dimensions are set to 0 as we are only interested in exerting force
  ↪  along the y-axis.
  - `arm_position` and `arm_velocity`: These inputs provide the current joint positions and velocities of
  ↪  the robot arm, respectively. They are directly obtained from `measured_arm_position` and
  ↪  `measured_arm_velocity`.

- **Output Ports:**
  - `applied_arm_torque`: This port outputs the computed joint torques for the robot arm, which are
  ↪  derived based on the desired force and pose control objectives.

### Constant Parameters Used
1. For the `LQRController`, we introduced a state cost matrix `Q` with higher weights on the pole's angle
↪  and angular velocity to emphasize the importance of stabilizing the pole. A relatively small control
↪  cost `R` is chosen to allow the application of necessary control efforts without being overly
↪  penalized.

2. In the `PoseForceController`, we used a constant target pose for the end-effector, with force control
↪  primarily along the y-axis. The Boolean array `force_control_axis_mask` is configured to apply force
↪  control only along the y-axis and pose control for all other dimensions.

### Summary
To connect these controllers effectively:
- The `state` input port of the `LQRController` must be assembled from available measurements,
↪  particularly focusing on the cart position and velocity along the y-axis and the pole's angle and
↪  angular velocity.
- The output control force from the `LQRController` must be properly directed to the `ee_force_target`
↪  input port of the `PoseForceController`, specifically into the dimension controlling force along the
↪  y-axis.
- The `ee_pose_target`, `arm_position`, and `arm_velocity` ports of the `PoseForceController` require
↪  direct measurements or set targets, with the end-effector pose being a specified constant.
- The final output of `applied_arm_torque` from the `PoseForceController` determines the joint torques to
↪  achieve both stabilization of the pole and maintaining the end-effector's desired pose.

This approach integrates both controllers to manage the primary objective of pole stabilization and the
↪  secondary objective of maintaining the end-effector pose efficiently.

## C.1.2 Composition via Semantic Understanding

## Meta-Control
The GPT is a proficient programmer of Python and PyDrake and a professor in control. The GPT will help the
↪  user to accomplish the code after reading the requirements. The GPT will be instructed step by step so
↪  please only complete mentioned tasks. The code implements a robot skill through a hierarchical design.
↪  A task controller is chosen to give high-level control, and a tracking controller is given to track
↪  the task control. The task controller and the tracking controller will be given. The GPT needs to
↪  accomplish a task controller converter and a tracking controller converter. A converter transforms
↪  available measurements and ports to the required input ports of the controller.

The GPT will proceed with the following steps: task_callback, and tracking_callback. In the GPT's
↪ response, the most important part will be <step_name> followed by a code block enclosed by
↪ ```python```. Each code block should correspond to a single step. This explicitly indicates the code
↪ is written for a specific step. The GPT can feel free to make some plans to help it think before
↪ writing the code blocks. There is no need to explain the code after the code block. The GPT should not
↪ import any module. The GPT should not make up arguments, variables, and anything else that is not
↪ mentioned.

## Meta-Control
Please help me to compose a skill to stabilize a cart pole. The cart is attached to the end effector of
↪ the robot arm. The pole rotates freely around the x-axis and is unactuated. Please stabilize the pole
↪ around the upright position (when the pole angle, denoted by PolePin, is 0) by exerting force on the
↪ cart along the y-axis. A secondary objective is to keep the end-effector pose as close as possible to
↪ [1.57078469e+00, 0.0, 1.57079455e+00, 5.96598901e-01, 1.34625984e-03, 4.33589236e-01] ([roll, pitch,
↪ yaw, x, y, z]). The cart is 0.1 kg, and the pole is 0.25 m, 0.01 kg.

## Assistant
Certainly! I'm good at writing callback functions for ports in PyDrake. I will follow your instructions
↪ carefully and think carefully.

## Meta-Control
Now please implement callback functions of a `LeafSystem` that acts as a connector which I have already
↪ constructed. The input ports correspond to inputs to the skill (observations and controller
↪ parameters). The output ports correspond to the input ports of the task controller.
Here is a yaml file describing what the inputs to the skill are:
{skill_input_summary}

Here we print the port name, type, and sample value for all the input ports. You can get the value of the
↪ port by `value = self.GetInputPort(port_name).Eval(context)` when implementing the call back
↪ functions:
{input_port_sample}

The chosen task model, tracking model, task controller, and tracking controller are
```python

x0 = np.zeros((4,1))  # Initializing the state for the cart-pole system
u0 = np.zeros((1,1))  # Initial control input (force)
m_cart = 0.1  # mass of the cart
m_pole = 0.01  # mass of the pole
l_pole = 0.25  # length of the pole
g = 9.81  # gravitational acceleration
# A and B matrices are placeholders and must be defined based on the specific dynamics of the cart-pole
↪ system
A = np.array([[0, 1, 0, 0],
              [0, 0, m_pole * g / m_cart, 0],
              [0, 0, 0, 1],
              [0, 0, (m_cart + m_pole) * g / (l_pole * m_cart), 0]])
B = np.array([[0], [1/m_cart], [0], [-1/(l_pole*m_cart)]])
C = np.eye(4)
D = np.zeros((4, 1))
task_model = setup_linear_model(x0, u0, A, B, C, D)

tracking_model = setup_arm_with_objects_model(arm_type, object_info)

Q = np.diag([1, 1, 10, 10])  # Emphasizing the importance of the pole's position and velocity
R = np.array([[0.1]])  # Control effort cost
task_controller = LQRController(task_model, Q, R, x0, u0)

force_control_axis_mask = np.array([False, False, False, False, True, False])  # Force control along the
↪ y-axis
tracking_controller = PoseForceController(tracking_model, force_control_axis_mask)

```

Here is the design summary to explain the expected way of how do the controllers work, and how to connect
↪ the ports:
{design_summary} (from Design via Templates)

Here is the summary of the input ports of the task controller:
{task_controller_input_summary} (from the controller summary)

The functions you need to implement are:
```python
def task_state(self, context, output: BasicVector)
    pass
```

```
Here is an example response:

<task_callback>
```python
def place_holder_function(self, context, output: BasicVector):
    abstract_input_port_value = self.GetInputPort(port_name).Eval(context)
    output.SetFromVector(abstract_input_port_value["place_holder_key"])
def place_holder_function2(self, context, output: BasicVector):
    vector_input_port_value = self.GetInputPort(port_name).Eval(context)
    out = np.zeros(6)
    out[3] = vector_input_port_value[0]
    output.SetFromVector(out)
def place_holder_function3(self, context, output: AbstractValue):
    value = self.GetInputPort(port_name).Eval(context)
    output.set_value(value)
```

Please implement all the task callback functions in one code block beginning with <task_callback>.

Here are some tips to implement the callback functions:
* When use a port, make sure the name of the port is correct and the port is available in the current
↪ function.
* Note that some of the ports should be passed through without any modification like control_parameters.
↪ Some of the ports need to be modified.
* Make sure the dimensions of the output match the port. The output value may be only partially available
↪ from the input port. You need to decide the value for the remaining. Be careful of which dimension to
↪ set value, make sure it complies with the description of the port in the controller summary.
* For pose, velocity, and force vectors, rotation always comes first, such as [roll, yaw, pitch, x, y, z]
↪ or [x-rotation, y-rotation, z-rotation, x-translation, y-translation, z-translation].
* If the input port is a dictionary, make sure only use keys that has shown in the printed port value
↪ sample.
* If the output should be an enum type, you can use the enum type mentioned in the controller summary
↪ directly without any import.
* Recall the task requirements given in the beginning. Make sure the call back functions correctly to
↪ solve the task.
* Please avoid using placeholders or simplified examples that require modifications by others. Ensure all
↪ calculations are conclusive. In case of any uncertainties about parameters, utilize the information
↪ provided to derive the most informed inference available.
* You can use `self.num_q` to denote the number of joints

Please implement the call-back functions. Please think step by step then write the code.

## Assistant
<task_callback>
```python
def task_state(self, context, output: BasicVector):
    # Extracting necessary inputs
    measured_body_poses = self.GetInputPort("measured_body_poses").Eval(context)
    measured_body_velocities = self.GetInputPort("measured_body_velocities").Eval(context)
    measured_joint_states = self.GetInputPort("measured_joint_states").Eval(context)

    # Initialize state vector with zeros
    state = np.zeros(4)

    # Position and Velocity of Cart along y-axis from body poses and velocities
    cart_pose = measured_body_poses['Cart'].translation()
    cart_velocity = measured_body_velocities['Cart'].translational()
    state[0] = cart_pose[1]   # y position of Cart
    state[1] = cart_velocity[1]   # y velocity of Cart

    # Pole's angle (around x-axis) and angular velocity from joint states
    pole_angle = measured_joint_states['PolePin']['positions']
    pole_angular_velocity = measured_joint_states['PolePin']['velocities']
    state[2] = pole_angle[0]   # Assuming 'positions' is a numpy array with one element
    state[3] = pole_angular_velocity[0]   # Assuming 'velocities' is a numpy array with one element

    output.SetFromVector(state)
```

## Meta-Control
Now please implement callback functions of a `LeafSystem` that acts as a connector which I have already
↪ constructed for the tracking controller. The input ports correspond to inputs to the skill
↪ (observations and controller parameters) and task control (the output of the task controller). The
↪ output ports correspond to the input ports of the tracking controller.

Recall the input port of the skill given previously, and the following is a description of the task
↪ control port. You can use all skill input ports and the task control port.
{task_control_port_summary} (from the controller summary)

Here is the summary of the input ports of the tracking controller:
```

{tracking_controller_input_summary} (from the controller summary)

The functions you need to implement are:
```python
def tracking_ee_pose_target(self, context, output: BasicVector)
    pass

def tracking_ee_force_target(self, context, output: BasicVector)
    pass

def tracking_arm_position(self, context, output: BasicVector)
    pass

def tracking_arm_velocity(self, context, output: BasicVector)
    pass

```

Here is an example response:

<tracking_callback>
```python
def place_holder_function(self, context, output: BasicVector):
    abstract_input_port_value = self.place_holder_port.Eval(context)
    output.SetFromVector(abstract_input_port_value["place_holder_key"])
def place_holder_function2(self, context, output: BasicVector):
    vector_input_port_value = self.place_holder_port.Eval(context)
    out = np.zeros(6)
    out[3] = vector_input_port_value[0]
    output.SetFromVector(out)
def place_holder_function3(self, context, output: AbstractValue):
    value = self.place_holder_port.Eval(context)
    output.set_value(value)
```

Follow the previously given tips for implementing the callback functions. Furthermore:
* Note that output ports of the task controller have been renamed by adding the prefix: 'task_' +
↪   port_name.

Now please implement the call-back functions. Please think step by step then write the code.

## Assistant
<tracking_callback>
```python
def tracking_ee_pose_target(self, context, output: BasicVector):
    # Desired end-effector pose is constant as per design summary
    desired_ee_pose = [1.57078469e+00, 0.0, 1.57079455e+00, 5.96598901e-01, 1.34625984e-03,
    ↪   4.33589236e-01]
    output.SetFromVector(np.array(desired_ee_pose))

def tracking_ee_force_target(self, context, output: BasicVector):
    # Obtaining the control action from task controller, directing it to apply force along y-axis
    task_control = self.GetInputPort("task_control").Eval(context)
    # Only force along the y-axis is controlled, the rest is masked out with zeros.
    ee_force_target = np.array([0.0, 0.0, 0.0, 0.0, task_control[0], 0.0])
    output.SetFromVector(ee_force_target)

def tracking_arm_position(self, context, output: BasicVector):
    # Directly passing the robot arm's measured position
    measured_arm_position = self.GetInputPort("measured_arm_position").Eval(context)
    output.SetFromVector(measured_arm_position)

def tracking_arm_velocity(self, context, output: BasicVector):
    # Directly passing the robot arm's measured velocity
    measured_arm_velocity = self.GetInputPort("measured_arm_velocity").Eval(context)
    output.SetFromVector(measured_arm_velocity)
```

## Meta-Control
Here is the output of running the skill: <empty>

Human expert supplementary instructions: <empty>

Do you think it is working as expected? (Empty output means the system is running without errors)
If it is, please answer with one word "yes".
If it is not, please identify which step(s) went wrong and explain how to fix it by including which
↪   step(s) to modify (<step_name>) and the updated code (enclosed by ```python```). We have proceeded
↪   with the following steps:
- task_callback
- tracking_callback

Please do not make up a nonexist step.

If the system is not working as expected, you must modify at least one step. You can modify multiple
↪  steps.
For each step, you can only modify the function that you think is wrong. Others will be kept unchanged.
Remember to add "<step_name>" before the corresponding code block. A correct solution exists with the
↪  given information. Try your best.

Here is a sample response when it is not working as expected:

I think this error occurred because [some reason].

updated_code:
<tracking_callback>

```python
def tracking_arm_position(self, context, output: BasicVector):
    ...
```

## Assistant
yes

## Meta-Control
Please summarize the task_controller, tracking_controll, task_callback and the tracking_callback. Explain
↪  the data flow. This response will be used as a prompt to choose the performance metric from
↪  measurements and tune the controller parameters, so please include all necessary information.

## Assistant
The integration of the task and tracking controllers with their respective callback functions forms a
↪  comprehensive control loop for stabilizing a cart-pole system attached to the end effector of a
↪  robotic arm. Here's a summary of the components and their data flow:

### Task Controller (`LQRController`)
- **Purpose:** Stabilizes the cart-pole system around the upright position by computing a control force to
↪  be applied along the y-axis of the cart.
- **Input Port:**
  - `state`: Accepts a state vector `[x, x_dot, theta, theta_dot]`, representing cart position and
  ↪  velocity along the y-axis, and pole angle and angular velocity, respectively.
- **Output Port:**
  - `control`: Outputs a control action (force) based on the LQR control law.
- **Implemented by Function:** `task_state` in `<task_callback>`.

### Tracking Controller (`PoseForceController`)
- **Purpose:** Aligns the end effector's pose with a target pose while applying the computed force from
↪  the task controller to stabilize the pole.
- **Input Ports:**
  - `ee_pose_target`: Takes a constant target pose for the end effector.
  - `ee_force_target`: Receives the control action (force along the y-axis) from the task controller to be
  ↪  applied by the end effector.
  - `arm_position`: Provides current joint positions of the robot arm.
  - `arm_velocity`: Provides current joint velocities of the robot arm.
- **Output Port:**
  - N/A (As these inputs directly dictate the behavior of the end effector and arm based on internal
  ↪  tracking control logic).
- **Implemented by Functions:** `tracking_ee_pose_target`, `tracking_ee_force_target`,
↪  `tracking_arm_position`, and `tracking_arm_velocity` in `<tracking_callback>`.

### Data Flow
1. **Observations to Task Controller:**
   - The cart's position and velocity, along with the pole angle and angular rate, are extracted from the
   ↪  `measured_body_poses`, `measured_body_velocities`, and `measured_joint_states`. These values are
   ↪  aggregated into a state vector `[x, x_dot, theta, theta_dot]` by the `task_state` function in the
   ↪  task controller callback, which feeds into the `LQRController`.

2. **Task Controller to Tracking Controller:**
   - The computed control action (force) from the `LQRController` is passed to the
   ↪  `tracking_ee_force_target` function in the tracking controller callback. It sets this force as the
   ↪  end effector's desired force along the y-axis while keeping forces in other dimensions at zero.

3. **Observations to Tracking Controller:**
   - Direct observations of the robot arm's joint positions and velocities (`measured_arm_position` and
   ↪  `measured_arm_velocity`) are passed through `tracking_arm_position` and `tracking_arm_velocity`
   ↪  functions to their respective input ports on the tracking controller.

4. **Constant Parameters:**
   - A constant target pose for the end effector is set within the `tracking_ee_pose_target` function,
   ↪  reflecting the secondary objective of maintaining the end-effector pose close to a predefined pose.

#### Performance Metric Selection and Controller Tuning:

To tune the controller parameters and select an appropriate performance metric, the complete flow of data
↪ and objectives should be considered:
- For the **task controller**, the performance metric might include the Euclidean distance between the
↪ current pole position/angle and its upright position (zero angle), the root-mean-square error (RMSE)
↪ of the pole's angle over time, or the control effort exerted.
- Tuning parameters for the task controller (`LQRController`) include adjusting the `Q` and `R` matrices,
↪ which balance between penalizing deviation from the target state and the magnitude of the control
↪ effort.

- For the **tracking controller**, the performance metric might focus on the precision of the end
↪ effector's pose alignment with its target and the magnitude of applied forces.
- Tuning involves the force control axis mask and potentially adjusting internal pose control parameters
↪ within the `PoseForceController` to achieve desired tracking performance and responsiveness.

By carefully monitoring these performance metrics and adjusting controller parameters accordingly, the
↪ control schemes can be refined to optimize both the primary objective of stabilizing the pole and the
↪ secondary objective of maintaining the desired end-effector pose.

## C.1.3 Parameter Optimization

```
## Meta-Control
The GPT is a proficient programmer of Python and PyDrake and a professor in control. The GPT will help the
↪ user to revise the code after reading the requirements. The GPT will be instructed step by step so
↪ please only complete mentioned tasks. The code implements a robot skill through a hierarchical design.
↪ A task controller is chosen to give high-level control, and a tracking controller is given to track
↪ the task control. The task controller and the tracking controller are connected through callback
↪ functions. The GPT will help to tune the parameters and check variable assignments of the hierarchical
↪ control systems. A time series of the task controller state and control will be given to help you
↪ determine if the system is running as expected. Then GPT needs to decide if the time series is
↪ desired, and then identify parameters and variables that can affect the time series. In the end, The
↪ GPT needs to tune the parameters and variables through multiple rounds of interaction.

The GPT can update any of the following code blocks: <task_model>, <tracking_model>, <task_controller>,
↪ <tracking_controller>, <task_callback>, <tracking_callback>. Each time, the GPT needs to pick and
↪ rewrite one or more blocks to update the parameters or variable assignments. The parameters and
↪ variables can be anything defined in the code. Sometimes, it may require switching the order of
↪ dimensions to correct the implementation error.

In the GPT's response, the most important part will be <block_name> followed by a code block enclosed by
↪ ```python```. This explicitly indicates the code is written for a specific block. The GPT needs to
↪ give the complete updated code block that contains the revision. The GPT can feel free to make some
↪ plans to help it think before writing the code blocks. There is no need to explain the code after the
↪ code block. The GPT should not import any module. The GPT should not make up arguments, variables, and
↪ anything else that is not mentioned.

## Meta-Control
Please help me to compose a skill to stabilize a cart pole. The cart is attached to the end effector of
↪ the robot arm. The pole rotates freely around the x-axis and is unactuated. Please stabilize the pole
↪ around the upright position (when the pole angle, denoted by PolePin, is 0) by exerting force on the
↪ cart along the y-axis. A secondary objective is to keep the end-effector pose as close as possible to
↪ [1.57078469e+00, 0.0, 1.57079455e+00, 5.96598901e-01, 1.34625984e-03, 4.33589236e-01] ([roll, pitch,
↪ yaw, x, y, z]). The cart is 0.1 kg, and the pole is 0.25 m, 0.01 kg.

## Assistant
Certainly! I'm good at tuning parameters for control systems in PyDrake. I will follow your instructions
↪ carefully and think carefully.

## Meta-Control
Here is the summary of the control system to explain the expected way of how do the controllers work, and
↪ how the ports are connected:
{implementation_summary} (from Interface Alignment)

Here is a yaml file describing what the inputs to the skill are:
{skill_input_summary}

Here we print the port name, type, and sample value for all the input ports:
{input_port_sample}

Here are the detailed implementation of the task_model, tracking_model, task_controller,
↪ tracking_controller, task_callback (the callback functions for the task input port), and
↪ tracking_callback (callback functions for the tracking input ports).

=======
<task_model>:
```python

x0 = np.zeros((4,1))  # Initializing the state for the cart-pole system
u0 = np.zeros((1,1))  # Initial control input (force)
```

```python
m_cart = 0.1  # mass of the cart
m_pole = 0.01  # mass of the pole
l_pole = 0.25  # length of the pole
g = 9.81  # gravitational acceleration
# A and B matrices are placeholders and must be defined based on the specific dynamics of the cart-pole
↪  system
A = np.array([[0, 1, 0, 0],
              [0, 0, m_pole * g / m_cart, 0],
              [0, 0, 0, 1],
              [0, 0, (m_cart + m_pole) * g / (l_pole * m_cart), 0]])
B = np.array([[0], [1/m_cart], [0], [-1/(l_pole*m_cart)]])
C = np.eye(4)
D = np.zeros((4, 1))
task_model = setup_linear_model(x0, u0, A, B, C, D)

```

=======
<tracking_model>:
```python

tracking_model = setup_arm_with_objects_model(arm_type, object_info)

```

=======
<task_controller>:
```python

Q = np.diag([1, 1, 10, 10])  # Emphasizing the importance of the pole's position and velocity
R = np.array([[0.1]])  # Control effort cost
task_controller = LQRController(task_model, Q, R, x0, u0)

```

=======
<tracking_controller>:
```python

force_control_axis_mask = np.array([False, False, False, False, True, False])  # Force control along the
↪  y-axis
tracking_controller = PoseForceController(tracking_model, force_control_axis_mask)

```

=======
<task_callback>:
```python

def task_state(self, context, output: BasicVector):
    # Extracting necessary inputs
    measured_body_poses = self.GetInputPort("measured_body_poses").Eval(context)
    measured_body_velocities = self.GetInputPort("measured_body_velocities").Eval(context)
    measured_joint_states = self.GetInputPort("measured_joint_states").Eval(context)
    # Initialize state vector with zeros
    state = np.zeros(4)
    # Position and Velocity of Cart along the y-axis from body poses and velocities
    cart_pose = measured_body_poses['Cart'].translation()
    cart_velocity = measured_body_velocities['Cart'].translational()
    state[0] = cart_pose[1]  # y position of Cart
    state[1] = cart_velocity[1]  # y velocity of Cart
    # Pole's angle (around the x-axis) and angular velocity from joint states
    pole_angle = measured_joint_states['PolePin']['positions']
    pole_angular_velocity = measured_joint_states['PolePin']['velocities']
    state[2] = pole_angle[0]  # Assuming 'positions' is a numpy array with one element
    state[3] = pole_angular_velocity[0]  # Assuming 'velocities' is a numpy array with one element
    output.SetFromVector(state)

```

=======
<tracking_callback>:
```python

def tracking_ee_pose_target(self, context, output: BasicVector):
    # Desired end-effector pose is constant as per design summary
    desired_ee_pose = [1.57078469e+00, 0.0, 1.57079455e+00, 5.96598901e-01, 1.34625984e-03,
    ↪  4.33589236e-01]
    output.SetFromVector(np.array(desired_ee_pose))
def tracking_ee_force_target(self, context, output: BasicVector):
    # Obtaining the control action from task controller, directing it to apply force along y-axis
```

```
        task_control = self.GetInputPort("task_control").Eval(context)
        # Only force along the y-axis is controlled, the rest is masked out with zeros.
        ee_force_target = np.array([0.0, 0.0, 0.0, 0.0, task_control[0], 0.0])
        output.SetFromVector(ee_force_target)
    def tracking_arm_position(self, context, output: BasicVector):
        # Directly passing the robot arm's measured position
        measured_arm_position = self.GetInputPort("measured_arm_position").Eval(context)
        output.SetFromVector(measured_arm_position)
    def tracking_arm_velocity(self, context, output: BasicVector):
        # Directly passing the robot arm's measured velocity
        measured_arm_velocity = self.GetInputPort("measured_arm_velocity").Eval(context)
        output.SetFromVector(measured_arm_velocity)

```
```

Here are time series data of the task space state and task space control, please identify metrics from
↪  them and determine if the system is running as expected.
{time series of task state and task control}

Please first think step by step about the expected time series and determine if the current series is
↪  desired.
Then please analyze one by one if any of the following happen.

* Mismatch of order. The order of the state is inconsistent in model definition, controller definition, or
↪  callback function. Double-check if the order is consistent. For example, an error can be modeling the
↪  task space with [pos_a, vel_a, pos_b, vel_b], but composed the task state by [pos_b, vel_b, pos_a,
↪  vel_a] in task callbacks. In this case, the GPT needs to redefine the state, the model, or the
↪  callback function.
* Error in definition. Double-check if the assignment is correct for the model, the state, and the
↪  callback functions. Make sure the signs are correct, the numbers are desired, there is no missing
↪  assignments or careless mistakes. Especially pay attention to the dynamic model. Think step by step
↪  about the meaning of each element in the dynamic function.
* Messed up rotation and translation order. In pydrake, for pose, velocity, and force vectors, rotation
↪  always comes first, such as [roll, yaw, pitch, x, y, z] or [x-rotation, y-rotation, z-rotation,
↪  x-translation, y-translation, z-translation]
* Modeling is inaccurate, that is when the model involves estimated parameters and it turns out the
↪  estimation is inaccurate.
* The controller parameter is not optimal to finish the task. For example, the system diverges with the
↪  current controller gain.

If the GPT thinks any one of these happens, then identify the parameters and variable assignments that
↪  require tuning to correct errors and improve the performance. The identified code blocks that contain
↪  these parameters and variables will be updated in later conversations.

## Assistant
Analyzing the provided system setup, time series data for the task state and control, and the goals and
↪  structure of the control system, let's break down the evaluation and identification process step by
↪  step.

### Analysis of Desired Time Series
The time series of **task_state** shows the cart-pole system's state vector components over time: cart
↪  position and velocity along the y-axis, and pole angle and angular velocity. The **task_control**
↪  shows the computed control actions (force) to be applied along the y-axis of the cart to stabilize the
↪  pole around the upright position.
- The pole angle (`theta`) initially decreases towards 0, indicating movement towards the upright
↪  position, which is desired. However, the absolute value of `theta` should ideally approach zero more
↪  closely over time, stabilizing around the upright position.
- The **task_control** force values start positive, then decrease, and become negative, suggesting efforts
↪  to stabilize the pole by moving the cart back and forth. The force values appear to reduce in
↪  magnitude over time, which might indicate a diminishing need for correction as the system stabilizes,
↪  a desirable trend.

### Expected and Actual Time Series Comparison
The time series appears to follow expected trends: corrections decrease as the system approaches a stable
↪  state. However, for precise stability and optimal control, further refined tuning of controller
↪  parameters and system modeling accuracy might be necessary.

### Identification of Potential Issues
1. **Mismatch of Order**: The definitions provided seem consistent in terms of order, especially in the
↪  `task_state` callback, where `[x, x_dot, theta, theta_dot]` order matches with how the system model
↪  and controller are defined.

2. **Error in Definition**: Initial checks don't reveal obvious errors in sign or incorrect assignments,
↪  but deeper evaluation is needed when examining specific code blocks.

3. **Messed Up Rotation and Translation Order**: Given the problem scope, it's primarily about linear
↪  motion and rotation around a single axis. The task controller deals with linear motion and doesn't
↪  directly manipulate rotations, while the tracking controller's force application is aligned correctly
↪  along the y-axis.
```

4. **Modeling Inaccuracy**: Based on the provided dynamics and control model, the fundamental modeling
↪  approach seems aligned with the physical behavior of a cart-pole system. However, specific parameters
↪  (masses, lengths) are accurate as provided and need no immediate correction.

5. **Controller Parameter Not Optimal**: Parameters within `Q` and `R` in the <task_controller> and force
↪  control axis mask in <tracking_controller> suggest room for optimization. For improved performance,
↪  especially in task prioritization (pole stability vs. cart position) and control effort, adjustments
↪  to these values may offer better results.

### Preliminary Recommendations for Parameters and Variable Assignments Tuning
To potentially correct errors and improve system performance, adjustments to `Q` and `R` matrix values in
↪  **<task_controller>** should be considered to better balance between penalizing the pole's deviation
↪  from the upright position and the energy (force) expended for control. The initial emphasis is on the
↪  pole's position (`theta`) and velocity (`theta_dot`), hinting at the right direction, but fine-tuning
↪  will help achieve a more precise and energy-efficient stabilization.

Next steps in tuning will be directed towards revising `<task_controller>` parameters with a focus on
↪  likely adjusting `Q` for a higher emphasis on reducing the pole's angle deviation and possibly
↪  refining `R` to manage control effort more effectively.

## Meta-Control
Now please decide if the time series is desired.
If it is, please answer with one word "yes".
If not, you must rewrite the code blocks that contain parameters and variables that require tuning to
↪  correct errors or to improve performance. Do not fear to make mistakes. We will evaluate the code
↪  block, produce a new time series, and iteratively improve the parameters.

The response must start with <step_name> followed by a code block enclosed by ```python```. <step_name>
↪  must be included before each code block so the code block can be recognized. step_name must be
↪  enclosed in angle brackets <>. <step_name> must be one of <task_model>, <tracking_model>,
↪  <task_controller>, <tracking_controller>, <task_callback>, <tracking_callback>. For <task_callback>,
↪  <tracking_callback>, you can only rewrite chosen functions of the codeblock. But for other steps, make
↪  sure to rewrite the whole code block because we will directly replace the original block with the
↪  rewritten one. Do not miss any variable in the code block.  During "rewriting", you are only supposed
↪  to modify the existing content of the code block, such as values assigned to parameters and variables.
↪  Do not revise functions and variables not presented in the existing code block.

An example output is:

<tracking_callback>
```python
def tracking_func2(self, context, output):
    # updated function body
```

<task_controller>
```python
# updated whole code block
```

## Assistant
The time series is not entirely desired. While it shows that the pole's angle is moving towards
↪  stabilization around the upright position, the control effort depicted by task_control is exhibiting a
↪  pattern of diminishing force application even when the pole has not reached a stabilized state (angle
↪  reaches and crosses 0 but starts to deviate again). The task_control should ideally drive the system
↪  more precisely towards stabilization without such a crossover. The adjustments needed seem to be in
↪  the tuning of the LQR controller's Q and R matrices, which directly affect the control performance by
↪  balancing between state deviations and the cost of control effort.

<task_controller>
```python
Q = np.diag([1, 1, 100, 10])  # Increased emphasis on the pole's angle stabilization
R = np.array([[0.01]])  # Reduced cost for control effort to allow more aggressive control actions
task_controller = LQRController(task_model, Q, R, x0, u0)
```

## Meta-Control
Here is the time series data of the task space state and task space control for the current code:
{}

Now please decide if the time series is desired.
If it is, please answer with one word "yes".
If not, you must rewrite the code blocks that contain parameters and variables that require tuning to
↪  correct errors or to improve performance. Do not fear to make mistakes. We will evaluate the code
↪  block, produce a new time series, and iteratively improve the parameters.

The response must start with <step_name> followed by a code block enclosed by ```python```. <step_name>
↪  must be included before each code block so the code block can be recognized. step_name must be
↪  enclosed in angle brackets <>. <step_name> must be one of <task_model>, <tracking_model>,
↪  <task_controller>, <tracking_controller>, <task_callback>, <tracking_callback>. For <task_callback>,
↪  <tracking_callback>, you can only rewrite chosen functions of the codeblock. But for other steps, make
↪  sure to rewrite the whole code block because we will directly replace the original block with the
↪  rewritten one. Do not miss any variable in the code block.  During "rewriting", you are only supposed
↪  to modify the existing content of the code block, such as values assigned to parameters and variables.
↪  Do not revise functions and variables not presented in the existing code block.

An example output is:

<tracking_callback>
```python
def tracking_func2(self, context, output):
    # updated function body
```

<task_controller>
```python
# updated whole code block
```

## Assistant
yes

---

## C.2   Pick and Place

## C.2.1   Design via Templates

---

## System
The GPT is a professor in robot control and a proficient programmer of Python and PyDrake. A robot task
↪  can be accomplished by sequentially executing several skills. The GPT will help the user write code to
↪  compose one of the skills after reading the requirements. The GPT will be instructed step by step. A
↪  skill is accomplished hierarchically by a task controller and a tracking controller. The task
↪  controller is responsible for generating high-level and abstract control, and the tracking controller
↪  is responsible for tracking the high-level control and satisfying constraints.

Here are some optional principles to design the task controller and the tracking controller:
1. Design the task controller for the object of interest or end-effector, and design the tracking
↪  controller for the robot.
2. Design the task controller on a simplified or approximated dynamics, such as Cartesian space or an
↪  approximated linear model, and design the tracking controller on the full robot dynamics.
3. If the a skill involves multiple objectives, design the task controller to take care of the primary
↪  objective and design the tracking controller to take care of the rest objectives and constraints. The
↪  task controller and the tracking controller can both output joint torque.
4. Task control is often the position, velocity, acceleration or force in the Cartesian space. But in some
↪  cases, the task control is best to be joint torque.
5. The final output of the tracking controller must be joint torques.

The GPT will proceed the following steps: task_controller, tracking_controller, task_model,
↪  tracking_model. In the GPT's response, the most important part will be <step_name> followed by a code
↪  block enclosed by ```python```. Each code block should correspond to a single step. This explicitly
↪  indicates the code is written for a specific step. The GPT can feel free to make some plan to help it
↪  think before writing the code blocks. There is no need to explain the code after the code block. The
↪  GPT should not import any module. The GPT should not make up arguments, variables and anything else
↪  that is not mentioned.

## User
Please help me write code that is part of a robot skill for a robot arm to reach a goal position in
↪  Cartesian space with collision avoidance. This skill will be used to pick and place objects in a
↪  clutterd scene with obstacles. Goal positions will be provided in the `controller_parameters` port.

## Assistant
Certainly! I'm good at writing controllers for robots with PyDrake. I will follow your instructions
↪  carefully and think carefully.

## Meta-Control
Let's first take a look of all available measurements, dynamic models, and controllers.

Here are all available inputs to the skill:
{skill_inputs_sample}

Here are all available dynamic model setup functions in yaml format:
{dynamic_model_brief_str}

Here are all available controllers in yaml format:

```
{controller_brief_str}
```

====================================================

Based on the available measurements, models, and controllers, please think step by step to choose the
↪  models and controllers:
1. Decide the subject of the task controller, is it an object, the end effector, or something else.
2. Decide the desired task control to be applied on the subject, is it Cartesian position, Cartesian
↪  force, or something else.
3. Decide the task space model that can model the dynamics of the subject with the task control. When the
↪  analytical form of the dynamics model is needed, draw upon your extensive knowledge in control theory
↪  and system modeling. Think step by step to make sure the model is correct according to the task
↪  description.
4. Decide the task controller that can apply the desired task control and realize the goal of the skill.
↪  Make sure the controller can realize the goal with the given available measurements. There is no
↪  additional inputs or functions provided to the controller.
5. Decide the tracking space model.
6. Decide the tracking controller to track the output of the task controller and to satisfy possible
↪  constraints.

Detail about how can the chosen models and controllers realize the skill goal. Be specific.

Please return the choice with the following format.

<task_model>
```python
task_model = setup_model(arg1, arg2)
```

<tracking_model>
```python
tracking_model = setup_model(arm_type, object_info)
```

<task_controller>
```python
task_controller = ControllerClass(arg1, arg2)
```

<tracking_controller>
```python
tracking_controller = ControllerClass(arg1)
```

Make sure your response comply with the following requirements:
* <step_name> must be included before each code block so the code block can be recognized. step_name must
↪  be enclosed in angle brackets <>.
* Choices have been made for all steps. A correct solution exists with the given infomation.
* Pass the arguments (args) when instantiating the controller and the model. The arguments needed are
↪  stated in the yaml file (args). **Do not** include items in "input_ports" (such as
↪  "controller_parameters") as arguments when instantiating the controllers. They are not arguments for
↪  instantiating the controller.
* The arguments should be defined before use, except `arm_type` and `object_info`, which can be used
↪  directly without definition.
* Do not use any placeholder or assume any variable is defined. All informations and available variables
↪  needed are provided. If your choice of model and controller requires additional information, rethink
↪  about it.
* The models and controllers are instantiated correctly as stated in the materials.
* The chosen model matches the required model of the controllers as stated in the materials.

## Assistant
Given the skill's goal for a robot arm to reach a goal position in Cartesian space with collision
↪  avoidance in a cluttered scene, let's derive the appropriate choices step by step.

1. The subject of the task controller will be the robot's end-effector since we need to move it to a
↪  specific position in the Cartesian space.
2. The desired task control to be applied on the subject is Cartesian position, aiming to move the end
↪  effector to the goal pose.
3. For the task space model, considering the requirements, we will use a model that concerns the robot arm
↪  and static objects in the scene to allow for consideration of obstacles for collision avoidance.
4. The task controller chosen aims to generate a trajectory that avoids collisions and reaches the goal
↪  position. Based on the available options, a Kinematic Trajectory Model Predictive Controller seems
↪  appropriate because it calculates a collision-free trajectory in joint space to achieve a goal in
↪  Cartesian space.
5. The tracking space model will be the model of the robot arm alone as the tracking controller's purpose
↪  is to follow the trajectory planned by the task controller.
6. The tracking controller should implement the following operation of the robot arm's actuators based on
↪  the task controller's output. A suitable choice might be a controller that directly receives joint
↪  positions and computes necessary torques, such as a JointStiffnessController.

Here's the detailed instantiation:

```
<task_model>
```python
task_model = setup_arm_with_objects_model(arm_type, object_info)
```

<tracking_model>
```python
tracking_model = setup_arm_model(arm_type)
```

<task_controller>
```python
task_controller = KinematicTrajectoryModelPredictiveController(model=task_model, resolve_period=3.0,
↪   num_steps=20)
```

<tracking_controller>
```python
tracking_controller = JointStiffnessController(model=tracking_model)
```
```

## Meta-Control
Here is the output of running the skill:

Human expert supplementary instructions:

Do you think it is working as expected? (Empty output means the system is running without errors)
If it is, please answer with one word "yes".
If it is not, please identify which step(s) went wrong and explain how to fix it by including which
↪   step(s) to modify (<step_name>) and the updated code (enclosed by ```python```). We have proceeded the
↪   following steps:
- task_model
- tracking_model
- task_controller
- tracking_controller

Please do not makeup an inexist step.

If the system is not working as expected, you must modify as least one step. You can modify multiple
↪   steps. Remember to add "<step_name>" before the corresponding code block. A correct solution exists
↪   with the given infomation. Try your best.

Here is a sample response when it is not working as expected:

I think this error occured because [some reason].

<tracking_controller>

updated_code:
```python
tracking_controller = NewTrackingController(arg1, arg2)
```

## Assistant
yes

## Meta-Control
Please summarize the choice of the task controller and the tracking controller. How would you expect the
↪   input output ports of the controllers will be connected to perform the skill. This response will be
↪   used as prompt to implement the port connector, so please include all necessary information,
↪   including:

* The detailed description of the input ports for the task controller and the tracking controller,
↪   including the meaning of each dimension (be careful about the order), and how to get them from the
↪   available measurements or outputs.
* A description of any constant parameter you have used that may affect the performance of the control
↪   system.
* Everthing else you think is necessary to implement the controller by only reading the summary.

## Assistant
For the task controller, we chose the **Kinematic Trajectory Model Predictive Controller**:

### Task Controller - Kinematic Trajectory Model Predictive Controller:

#### Input Ports:

1. **controller_parameters**: Comes from the skill inputs under the 'controller_parameters' port. It
   ↪ contains information about the desired goal pose for the end effector, expressed in the form of a
   ↪ `RigidTransform`, which specifies the target position and orientation in Cartesian space.

2. **measured_arm_position**: This input should be connected to the 'measured_arm_position' from the skill
   ↪ inputs, representing the current joint positions of the robot arm.

3. **measured_arm_velocity**: This input should be linked to the 'measured_arm_velocity' from the skill
   ↪ inputs, indicating the current joint velocities of the robot arm.

#### Notable Parameters:

- **resolve_period**: This parameter affects how often the MPC recalculates the trajectory. A lower value
   ↪ will make the trajectory more responsive to changes but will require more computational resources.

- **num_steps**: Determines the number of discrete steps within the control horizon of the MPC. A higher
   ↪ number of steps allows for a more finely detailed trajectory but increases computational demands.

### Tracking Controller - Joint Stiffness Controller:

#### Input Ports:

1. **controller_parameters**: While this specific tracking controller (Joint Stiffness Controller) does
   ↪ not inherently require `controller_parameters` as specified in its YAML interface, it generally
   ↪ accepts dynamic adjustments such as 'joint_stiffness' and 'joint_damping', which can be applied via an
   ↪ AbstractValue input if controller customization is necessary during runtime.

2. **arm_target_position**: Should be wired to the output of the task controller (Kinematic Trajectory
   ↪ Model Predictive Controller), which generates target joint positions the robot arm should follow to
   ↪ reach the desired end-effector pose in a collision-free manner.

3. **arm_position**: This is connected to 'measured_arm_position' from the skill measurements, which
   ↪ indicates the current positions of the robot arm's joints.

4. **arm_velocity**: This is connected to 'measured_arm_velocity' from the skill measurements, providing
   ↪ the current velocities of the robot arm's joints.

#### Notable Parameters:

Specific parameter settings like joint stiffness and damping coefficients (if they were to be dynamically
   ↪ adjustable through `controller_parameters`) affect how the robot arm physically behaves - its
   ↪ compliance and responsiveness to control signals.

### Implementation Details:

For effective control, the output port of the **Kinematic Trajectory Model Predictive Controller** (target
   ↪ joint positions) is connected to the input port for **arm_target_position** of the **Joint Stiffness
   ↪ Controller**. The measured arm positions and velocities, which are essential for both controllers,
   ↪ come directly from the robot's sensory feedback.

Additionally, the task controller's `controller_parameters` input is based on the skill's requirement
   ↪ (e.g., for collision avoidance and reaching the goal pose), which includes specifying the end-effector
   ↪ target pose that must be extracted from the 'controller_parameters' skill input.

This combination ensures that the robot follows a planned, collision-free trajectory to reach the target
   ↪ position in Cartesian space by adjusting its joint torques according to the deviations from the target
   ↪ joint positions computed by the task controller, considering current arm positions and velocities for
   ↪ accurate and efficient control.

---

## C.2.2  Composition via Semantic Understanding

---

## System
The GPT is a proficient programmer of Python and PyDrake and a professor in control. The GPT will help the
   ↪ user to accomplish the code after reading the requirements. The GPT will be instructed step by step so
   ↪ please only complete mentioned tasks. The code implements a robot skill through a hierarchical design.
   ↪ A task controller is chosen to give high-level control, and a tracking controller is given to track
   ↪ the task control. The task controller and the tracking controller will be given. The GPT needs to
   ↪ accomplish a task controller converter and a tracking controller converter. A converter transforms
   ↪ available measurements and ports to the required input ports of the controller.

The GPT will proceed the following steps: task_callback, tracking_callback. In the GPT's response, the
   ↪ most important part will be <step_name> followed by a code block enclosed by ```python```. Each code
   ↪ block should correspond to a single step. This explicitly indicates the code is written for a specific
   ↪ step. The GPT can feel free to make some plan to help it think before writing the code blocks. There
   ↪ is no need to explain the code after the code block. The GPT should not import any module. The GPT
   ↪ should not make up arguments, variables and anything else that is not mentioned.

Please help me write code that is part of a robot skill for a robot arm to reach a goal position in
↪   Cartesian space with collision avoidance. This skill will be used to pick and place objects in a
↪   clutterd scene with obstacles. Goal positions will be provided in the `controller_parameters` port.

## Assistant
Certainly! I'm good at writing callback functions for ports in PyDrake. I will follow your instructions
↪   carefully and think carefully.

## Meta-Control
Now please implement callback functions of a `LeafSystem` that acts as a connector which I have already
↪   constructed. The input ports correspond to inputs to the skill (observations and controller
↪   parameters). The output ports correspond to the input ports of the task controller.
Here is a yaml file describing what the inputs to the skill are:
{skill_input_summary}

Here we print the port name, type and sample value for all the input port. You can get the value of the
↪   port by `value = self.GetInputPort(port_name).Eval(context)` when impelementing the call back
↪   functions:
{input_port_sample}

The chosen task model, tracking model, task controller, and tracking controller are
```python

task_model = setup_arm_with_objects_model(arm_type, object_info)

tracking_model = setup_arm_model(arm_type)

task_controller = KinematicTrajectoryModelPredictiveController(model=task_model, resolve_period=3.0,
↪   num_steps=20)

tracking_controller = JointStiffnessController(model=tracking_model)

```

Here is the design summary to explain the expected way of how do the controllers works, and how to connect
↪   the ports:
{design_summary} (from Design via Templates)

Here is the summary of the input ports of the task controller:
{task_controller_input_summary} (from the controller summary)

The functions you need to implement are:
```python
def task_controller_parameters(self, context, output: AbstractValue)
    pass

def task_measured_arm_position(self, context, output: BasicVector)
    pass

def task_measured_arm_velocity(self, context, output: BasicVector)
    pass

```
Here is an example response:

<task_callback>
```python
def place_holder_function(self, context, output: BasicVector):
    abstract_input_port_value = self.GetInputPort(port_name).Eval(context)
    output.SetFromVector(abstract_input_port_value["place_holder_key"])
def place_holder_function2(self, context, output: BasicVector):
    vector_input_port_value = self.GetInputPort(port_name).Eval(context)
    out = np.zeros(6)
    out[3] = vector_input_port_value[0]
    output.SetFromVector(out)
def place_holder_function3(self, context, output: AbstractValue):
    value = self.GetInputPort(port_name).Eval(context)
    output.set_value(value)
```
Please implement all the task callback functions in one code block begin with <task_callback>.

Here are some tips to implement the callback functions:
* When use a port, make sure the name of the port is correct and the port is avialble in the current
↪   function.
* Note that some of the ports should be passed through without any modification like control_parameters.
↪   Some of the ports need to be modified.

* Make sure the dimensions of the output matches the port. The output value may be only partially
↪  available from input port. You need to decide the value for the remainings. Be careful of which
↪  dimension to set value, make sure it complies with the description of the port in the controller
↪  summary.
* For pose, velocity and force vectors, rotation always comes first, such as [roll, yaw, pitch, x, y, z]
↪  or [x-rotation, y-rotation, z-rotation, x-translation, y-translation, z-translation].
* If the input port is a dictionary, make sure only use keys that has shown in the printed port value
↪  sample.
* If the output should be a enum type, you can use the enum type mentioned in the controller summary
↪  directly without any import.
* Recall the task requirements given in the beginning. Make sure the call back functions correctly solve
↪  the task.
* Please avoid using placeholders or simplified examples that require modifications by others. Ensure all
↪  calculations are conclusive. In case of any uncertainties about parameters, utilize the information
↪  provided to derive the most informed inference available.
* You can use `self.num_q` to denote the number of joints

Please implement the call-back functions. Please think step by step then write the code.

## Assistant
<task_callback>
```python
def task_controller_parameters(self, context, output: AbstractValue):
    # Directly pass the controller_parameters from the input to the output.
    params = self.GetInputPort("controller_parameters").Eval(context)
    output.set_value(params)

def task_measured_arm_position(self, context, output: BasicVector):
    # Directly pass the measured_arm_position from the input to the output.
    measured_arm_position = self.GetInputPort("measured_arm_position").Eval(context)
    output.SetFromVector(measured_arm_position)

def task_measured_arm_velocity(self, context, output: BasicVector):
    # Directly pass the measured_arm_velocity from the input to the output.
    measured_arm_velocity = self.GetInputPort("measured_arm_velocity").Eval(context)
    output.SetFromVector(measured_arm_velocity)
```

## Meta-Control
Now please implement callback functions of a `LeafSystem` that acts as a connector which I have already
↪  constructed for the tracking controller. The input ports correspond to inputs to the skill
↪  (observations and controller parameters) and task control (the output of the task controller). The
↪  output ports correspond to the input ports of the tracking controller.

Recall the input port of the skill given previously, and the following is a description of the task
↪  control port. You can use all skill input ports and the task control port.
```yaml
- description: Target joint positions for the robot arm.
  name: task_q_nom
  size: num_q
  type: BasicVector

```

Here is the summary of the input ports of the tracking controller:
```yaml
- description: Input port for dynamic adjustment of controller parameters, such as
    'joint_stiffness' and 'joint_damping'.
  name: controller_parameters
  type: AbstractValue
- description: Target position for the arm.
  name: arm_target_position
  type: BasicVector
- description: Current position of the arm.
  name: arm_position
  type: BasicVector
- description: Current velocity of the arm.
  name: arm_velocity
  type: BasicVector

```

The functions you need to implement are:
```python
def tracking_controller_parameters(self, context, output: AbstractValue)
    pass

def tracking_arm_target_position(self, context, output: BasicVector)
    pass
```

```python
    def tracking_arm_position(self, context, output: BasicVector)
        pass

    def tracking_arm_velocity(self, context, output: BasicVector)
        pass
```

Here is an example response:

<tracking_callback>
```python
def place_holder_function(self, context, output: BasicVector):
    abstract_input_port_value = self.place_holder_port.Eval(context)
    output.SetFromVector(abstract_input_port_value["place_holder_key"])
def place_holder_function2(self, context, output: BasicVector):
    vector_input_port_value = self.place_holder_port.Eval(context)
    out = np.zeros(6)
    out[3] = vector_input_port_value[0]
    output.SetFromVector(out)
def place_holder_function3(self, context, output: AbstractValue):
    value = self.place_holder_port.Eval(context)
    output.set_value(value)
```

Follow the previously given tips for implementing the callback functions. Further more:
* Note that output ports of the task controller have been renamed by adding prefix: 'task_' + port_name.

Now please implement the call-back functions. Please think step by step then write the code.

## Assistant
<tracking_callback>
```python
def tracking_controller_parameters(self, context, output: AbstractValue):
    # Since the specific tracking controller (Joint Stiffness Controller) might not inherently require
    # `controller_parameters` in the same way the task controller does for its primary function,
    # we can decide to either pass a default configuration or dynamically adjust the parameters if needed.
    # Here, we will dynamically adjust the parameters based on the skill input.
    controller_parameters = self.GetInputPort("controller_parameters").Eval(context)
    output.set_value(controller_parameters)

def tracking_arm_target_position(self, context, output: BasicVector):
    # This should be wired to the output of the task controller, which provides target joint positions.
    task_q_nom = self.GetInputPort("task_q_nom").Eval(context)
    output.SetFromVector(task_q_nom)

def tracking_arm_position(self, context, output: BasicVector):
    # Pass the measured_arm_position from the input to the output.
    measured_arm_position = self.GetInputPort("measured_arm_position").Eval(context)
    output.SetFromVector(measured_arm_position)

def tracking_arm_velocity(self, context, output: BasicVector):
    # Pass the measured_arm_velocity from the input to the output.
    measured_arm_velocity = self.GetInputPort("measured_arm_velocity").Eval(context)
    output.SetFromVector(measured_arm_velocity)
```

## Meta-Control
Here is the output of running the skill:

Reached the cube! Switch to stage 1

Grasped the cube! Switch to stage 2

Human expert supplementary instructions:

Do you think it is working as expected? (Empty output means the system is running without errors)
If it is, please answer with one word "yes".
If it is not, please identify which step(s) went wrong and explain how to fix it by including which
↪ step(s) to modify (<step_name>) and the updated code (enclosed by ```python```). We have proceeded the
↪ following steps:
- task_callback
- tracking_callback

Please do not makeup an inexist step.

If the system is not working as expected, you must modify as least one step. You can modify multiple
↪ steps.

For each step, you can only modify the function that you think is wrong. Others will be kept unchanged.
Remember to add "<step_name>" before the corresponding code block. A correct solution exists with the
↪ given infomation. Try your best.

Here is a sample response when it is not working as expected:

I think this error occured because [some reason].

updated_code:
<tracking_callback>

```python
def tracking_arm_position(self, context, output: BasicVector):
    ...
```

## Assistant
yes

## Meta-Control
Please summarize the task_controller, tracking_controll, task_callback and the tracking_callback. Explain
↪ the data flow. This response will be used as prompt to choose the performance metric from
↪ measurements, and tune the controller parameters, so please include all necessary information.

## Assistant
The robot skill implemented involves controlling a robot arm to reach a goal position in Cartesian space
↪ while avoiding collisions, which is particularly useful for tasks like pick and place in cluttered
↪ scenes. To achieve this, a hierarchical control approach was adopted, consisting of a task controller
↪ and a tracking controller, interconnected through callback functions that serve as data converters.

### Task Controller

**Type:** Kinematic Trajectory Model Predictive Controller (KinematicTrajectoryModelPredictiveController)

**Role:** This controller is responsible for high-level decision-making. It calculates collision-free
↪ trajectories to reach the target end-effector pose, given as a goal in the Cartesian space.

**Input Ports:**
1. **controller_parameters:** Receives dynamic controller parameters such as the desired goal pose for the
↪ end effector (expressed as a `RigidTransform`, specifying target position and orientation) and
↪ possible gripper commands.
2. **measured_arm_position:** Receives the current joint positions of the robot arm.
3. **measured_arm_velocity:** Receives the current joint velocities of the robot arm.

**Notable Parameters:**
- **resolve_period:** Determines how often the controller recalculates the trajectory.
- **num_steps:** The number of steps in the control horizon of the MPC, influencing the granularity of the
↪ planned trajectory.

### Tracking Controller

**Type:** Joint Stiffness Controller (JointStiffnessController)

**Role:** This controller deals with lower-level control, ensuring that the robot's joints follow the
↪ target positions generated by the task controller to execute the trajectory accurately.

**Input Ports:**
1. **controller_parameters:** If necessary, receives dynamic adjustments such as 'joint_stiffness' and
↪ 'joint_damping'.
2. **arm_target_position:** Receives target joint positions to follow, as computed by the task controller.
3. **arm_position:** Connected to the robot's sensory feedback to obtain the current joint positions.
4. **arm_velocity:** Also connected to the robot's sensory feedback for current joint velocities.

### task_callback

This set of callback functions prepares the data from the robot's sensory inputs and controller parameters
↪ for the task controller. This includes:
- **task_controller_parameters:** Passes the `controller_parameters` directly to the task controller.
- **task_measured_arm_position:** Directly forwards the `measured_arm_position` to the task controller.
- **task_measured_arm_velocity:** Directly forwards the `measured_arm_velocity` to the task controller.

### tracking_callback

This set of callback functions converts outputs from the task controller and sensory inputs into the
↪ format required by the tracking controller. This includes:
- **tracking_controller_parameters:** Optionally passes dynamic controller adjustments to the tracking
↪ controller.
- **tracking_arm_target_position:** Forwards the target joint positions from the task controller to the
↪ tracking controller.

```
- **tracking_arm_position:** Passes the current joint positions from sensory feedback to the tracking
↪  controller.
- **tracking_arm_velocity:** Passes the current joint velocities also from sensory feedback to the
↪  tracking controller.

### Data Flow

The data flow begins with sensory inputs and dynamic parameters being fed into the task controller via the
↪  callback functions designed in `task_callback`. The task controller uses this data to compute a
↪  collision-free trajectory towards the goal pose. The target joint positions generated by the task
↪  controller are then passed to the tracking controller through `tracking_callback`. Additionally, the
↪  current joint states are also provided to the tracking controller, enabling it to compute how to move
↪  the robot's joints to follow the planned trajectory accurately.

This hierarchical and modular design allows for high-level path planning and low-level execution to be
↪  optimized individually, while the callback functions ensure they are compatible and can work together
↪  seamlessly.
```

## C.3 Open Door

## C.3.1 Design via Templates

```
## System
The GPT is a professor in robot control and a proficient programmer of Python and PyDrake. A robot task
↪  can be accomplished by sequentially executing several skills. The GPT will help the user write code to
↪  compose one of the skills after reading the requirements. The GPT will be instructed step by step. A
↪  skill is accomplished hierarchically by a task controller and a tracking controller. The task
↪  controller is responsible for generating high-level and abstract control, and the tracking controller
↪  is responsible for tracking the high-level control and satisfying constraints.

Here are some optional principles to design the task controller and the tracking controller:
1. Design the task controller for the object of interest or end-effector, and design the tracking
↪  controller for the robot.
2. Design the task controller on a simplified or approximated dynamics, such as Cartesian space or an
↪  approximated linear model, and design the tracking controller on the full robot dynamics.
3. If the a skill involves multiple objectives, design the task controller to take care of the primary
↪  objective and design the tracking controller to take care of the rest objectives and constraints. The
↪  task controller and the tracking controller can both output joint torque.
4. Task control is often the position, velocity, acceleration or force in the Cartesian space. But in some
↪  cases, the task control is best to be joint torque.
5. The final output of the tracking controller must be joint torques.

The GPT will proceed the following steps: task_controller, tracking_controller, task_model,
↪  tracking_model. In the GPT's response, the most important part will be <step_name> followed by a code
↪  block enclosed by ```python```. Each code block should correspond to a single step. This explicitly
↪  indicates the code is written for a specific step. The GPT can feel free to make some plan to help it
↪  think before writing the code blocks. There is no need to explain the code after the code block. The
↪  GPT should not import any module. The GPT should not make up arguments, variables and anything else
↪  that is not mentioned.

## User
Please help me write code for a robot skill designed to pull open a door.
Coordinate System Orientation:
The x-axis points forward.
The y-axis points to the left.
The z-axis points upward.
Currently, the door is closed, and the robot has successfully grasped the handle. When viewed from above,
↪  the robot base is at [0, 0], the door handle's position is approximately [0.8, 0.2], the hinge
↪  (rotation axis) is positioned at [0.8, 0.2 - radius], with "radius" being the handle's rotation
↪  radius, which fluctuates between 0.3 and 0.5 meters, and the door opens by rotating anticlockwise
↪  around the hinge. Therefore the vector from the hinge to the handle is currently [0, radius]. The
↪  height of the door handle is 0.5, which should remain unchanged during the pulling process. Given the
↪  uncertainty in the radius, the skill should not only determine a rough trajectory to pull the door
↪  open approximately 90 degrees, but also be compliant to avoid damaging the door during the pulling
↪  process.

## Assistant
Certainly! I'm good at writing controllers for robots with PyDrake. I will follow your instructions
↪  carefully and think carefully.

## Meta-Control
Let's first take a look of all available measurements, dynamic models, and controllers.

Here are all available inputs to the skill:
{skill_inputs_sample}
```

Here are all available dynamic model setup functions in yaml format:
{dynamic_model_brief_str}

Here are all available controllers in yaml format:
{controller_brief_str}

=======================================================

Based on the available measurements, models, and controllers, please think step by step to choose the
↪  models and controllers:
1. Decide the subject of the task controller, is it an object, the end effector, or something else.
2. Decide the desired task control to be applied on the subject, is it Cartesian position, Cartesian
↪  force, or something else.
3. Decide the task space model that can model the dynamics of the subject with the task control. When the
↪  analytical form of the dynamics model is needed, draw upon your extensive knowledge in control theory
↪  and system modeling. Think step by step to make sure the model is correct according to the task
↪  description.
4. Decide the task controller that can apply the desired task control and realize the goal of the skill.
↪  Make sure the controller can realize the goal with the given available measurements. There is no
↪  additional inputs or functions provided to the controller.
5. Decide the tracking space model.
6. Decide the tracking controller to track the output of the task controller and to satisfy possible
↪  constraints.

Detail about how can the chosen models and controllers realize the skill goal. Be specific.

Please return the choice with the following format.

<task_model>
```python
task_model = setup_model(arg1, arg2)
```

<tracking_model>
```python
tracking_model = setup_model(arm_type, object_info)
```

<task_controller>
```python
task_controller = ControllerClass(arg1, arg2)
```

<tracking_controller>
```python
tracking_controller = ControllerClass(arg1)
```

Make sure your response comply with the following requirements:
* <step_name> must be included before each code block so the code block can be recognized. step_name must
↪  be enclosed in angle brackets <>.
* Choices have been made for all steps. A correct solution exists with the given infomation.
* Pass the arguments (args) when instantiating the controller and the model. The arguments needed are
↪  stated in the yaml file (args). **Do not** include items in "input_ports" (such as
↪  "controller_parameters") as arguments when instantiating the controllers. They are not arguments for
↪  instantiating the controller.
* The arguments should be defined before use, except `arm_type` and `object_info`, which can be used
↪  directly without definition.
* Do not use any placeholder or assume any variable is defined. All informations and available variables
↪  needed are provided. If your choice of model and controller requires additional information, rethink
↪  about it.
* The models and controllers are instantiated correctly as stated in the materials.
* The chosen model matches the required model of the controllers as stated in the materials.

## Assistant
For accomplishing a robot skill designed to pull open a door, let's follow the structured steps to select
↪  the appropriate models and controllers.

1. **Subject of the Task Controller**: The subject of the task controller will be the end-effector because
↪  the task involves pulling the door open. Given this action is highly dependent on the position and
↪  orientation of the end-effector (in this case, the robot hand holding the door handle), precise
↪  control over its Cartesian position is crucial.

2. **Desired Task Control**: The desired task control will be the Cartesian position. This choice is
↪  because the robot needs to maintain a certain trajectory to pull the door open effectively. The task
↪  involves moving in a specific path that approximates a segment of a circle (due to the door's rotation
↪  around its hinge) while keeping the height constant.

3. **Task Space Model**: Since the dynamics of moving the door handle can be simplified to movements in
   ↪ the Cartesian space (where the robot needs to follow a specific trajectory to pull open the door), a
   ↪ placeholder model suffices for the task space model because the specific dynamics aren't as relevant
   ↪ to computing the desired end-effector trajectory.

4. **Task Controller**: The CartesianInterpolationController is apt for the task because it can generate a
   ↪ trajectory for the end-effector to reach the target position, considering the need to pull the door.
   ↪ This controller can interpolate a path towards the given way points, aligning with the need to adjust
   ↪ the pulling trajectory according to the door handle's position and maintaining compliance.

5. **Tracking Space Model**: For the tracking model, since the robot's interaction with the environment
   ↪ (i.e., the door) also needs consideration, especially ensuring that the motion does not lead to
   ↪ excessive force which could damage the door, we should utilize the `setup_arm_with_objects_model`.
   ↪ This model allows for considering the robot arm and static objects in the scene, facilitating avoiding
   ↪ unnecessary force imposition on the door during pulling.

6. **Tracking Controller**: Given the requirements for compliance and potential adjustments in the pulling
   ↪ force to accommodate the door's opening motion and its varying radius, a CartesianStiffnessController
   ↪ is suitable. This controller can regulate forces exerted by the robot, making the interaction more
   ↪ compliant and safer, which aligns with the goals of being compliant and avoiding damage.

Let's instantiate the chosen models and controllers with the provided arguments.

<task_model>
```python
task_model = setup_placeholder_model()
```

<tracking_model>
```python
tracking_model = setup_arm_with_objects_model(arm_type, object_info)
```

<task_controller>
```python
task_controller = CartesianInterpolationController(model=task_model)
```

<tracking_controller>
```python
tracking_controller = CartesianStiffnessController(model=tracking_model)
```

## Meta-Control
Please summarize the choice of the task controller and the tracking controller. How would you expect the
↪ input output ports of the controllers will be connected to perform the skill. This response will be
↪ used as prompt to implement the port connector, so please include all necessary information,
↪ including:

* The detailed description of the input ports for the task controller and the tracking controller,
↪ including the meaning of each dimension (be careful about the order), and how to get them from the
↪ available measurements or outputs.
* A description of any constant parameter you have used that may affect the performance of the control
↪ system.
* Everthing else you think is necessary to implement the controller by only reading the summary.

## Assistant
To implement the skill of pulling open a door with our robot, we've selected a
↪ **CartesianInterpolationController** as the task controller and a **CartesianStiffnessController** as
↪ the tracking controller. Below is the detailed information on their input ports and how these
↪ controllers will be interconnected to achieve the desired skill.

### Task Controller: CartesianInterpolationController

**Input Ports and Their Sources:**

1. **controller_parameters**: This input contains parameters for trajectory generation like
   ↪ `new_trajectory` flag, `way_points` with `times` and `points`, and `execution_time`. These can be
   ↪ directly sourced from the `controller_parameters` provided in the skill inputs.

2. **measured_ee_pose**: Needs a pose of the end effector in a 6D vector `[roll, pitch, yaw, x, y, z]`.
   ↪ This comes directly from `measured_ee_pose` in the available measurements.

3. **execution_time**: This input is required to evaluate the current position along the interpolated
   ↪ trajectory. It should be calculated based on the current time within the execution of a trajectory and
   ↪ could require a separate tracking mechanism or utilization of system/skill execution time metadata.

4. **target_position**: The final target position `[X, Y, Z]` for the end effector. Initially, this is
   ↪ derived from controller parameters under `way_points` or adjusted based on real-time measurements or
   ↪ conditions, including door hinge position and estimated radius.

**Output Port:**

- **ee_pose_nom**: This output provides the nominal (desired) end-effector pose `[roll, pitch, yaw, x, y,`
↪ `z]` based on the interpolation of the waypoints over the given times. This will be used as input to
↪ the tracking controller for actual motion execution.

### Tracking Controller: CartesianStiffnessController

**Input Ports and Their Sources:**

1. **ee_target (from task_controller's output)**: The desired end-effector target that includes both
↪ position and orientation in the form `[roll, pitch, yaw, x, y, z]`. This comes from the `ee_pose_nom`
↪ output port of our task controller.

2. **ee_target_type**: For this implementation, we need to specify that our target type is a pose
↪ (position + orientation), implying the controller should focus on tracking the pose rather than
↪ velocity or force. This is a constant parameter based on our control strategy to pull the door open by
↪ following a pose trajectory.

3. **arm_position**: Current joint positions of the robot arm, obtained from `measured_arm_position` in
↪ the available measurements.

4. **arm_velocity**: Current joint velocities of the robot arm, available from `measured_arm_velocity` in
↪ the measurements.

5. **controller_parameters**: Parameters such as `cartesian_stiffness` and `cartesian_damping` that define
↪ the compliance characteristics of the control. Values like `[50, 50, 50, 500, 500, 500]` for stiffness
↪ and `[5, 5, 5, 40, 40, 40]` for damping can be adopted from the `controller_parameters` in the skill
↪ inputs to suit the compliance and control behavior requirements.

**Output Port:**

- **applied_arm_torque**: The calculated joint torques to be applied to the robot arm to follow the
↪ desired trajectory while maintaining compliance with the imposed stiffness and damping parameters.

### Interconnecting the Controllers:

The output of the **CartesianInterpolationController** (`ee_pose_nom`) should be connected to the
↪ **CartesianStiffnessController**'s `ee_target` input port. This connection is crucial for the
↪ execution flow where the interpolated pose trajectory (desired behavior) informs the compliant motion
↪ control, resulting in the actual joint torques applied to the robot's actuators.

Parameters such as `ee_target_type` and `controller_parameters` for the **CartesianStiffnessController**
↪ must be carefully determined and set to ensure the robot's motion is both compliant (to prevent
↪ damage) and precise enough to accomplish the task of pulling the door open without requiring forceful
↪ interactions.

By carefully adjusting these parameters and ensuring accurate port connections, the robot should be able
↪ to pull the door open by following the desired trajectory, all while adjusting its actions based on
↪ real-time dynamics and constraints of the task environment.

---

## C.3.2  Composition via Semantic Understanding

---

## System
The GPT is a proficient programmer of Python and PyDrake and a professor in control. The GPT will help the
↪ user to accomplish the code after reading the requirements. The GPT will be instructed step by step so
↪ please only complete mentioned tasks. The code implements a robot skill through a hierarchical design.
↪ A task controller is chosen to give high-level control, and a tracking controller is given to track
↪ the task control. The task controller and the tracking controller will be given. The GPT needs to
↪ accomplish a task controller converter and a tracking controller converter. A converter transforms
↪ available measurements and ports to the required input ports of the controller.

The GPT will proceed the following steps: task_callback, tracking_callback. In the GPT's response, the
↪ most important part will be <step_name> followed by a code block enclosed by ```python```. Each code
↪ block should correspond to a single step. This explicitly indicates the code is written for a specific
↪ step. The GPT can feel free to make some plan to help it think before writing the code blocks. There
↪ is no need to explain the code after the code block. The GPT should not import any module. The GPT
↪ should not make up arguments, variables and anything else that is not mentioned.

## User
Please help me write code for a robot skill designed to pull open a door.
Coordinate System Orientation:
The x-axis points forward.
The y-axis points to the left.
The z-axis points upward.

Currently, the door is closed, and the robot has successfully grasped the handle. When viewed from above,
↪  the robot base is at [0, 0], the door handle's position is approximately [0.8, 0.2], the hinge
↪  (rotation axis) is positioned at [0.8, 0.2 - radius], with "radius" being the handle's rotation
↪  radius, which fluctuates between 0.3 and 0.5 meters, and the door opens by rotating anticlockwise
↪  around the hinge. Therefore the vector from the hinge to the handle is currently [0, radius]. The
↪  height of the door handle is 0.5, which should remain unchanged during the pulling process. Given the
↪  uncertainty in the radius, the skill should not only determine a rough trajectory to pull the door
↪  open approximately 90 degrees, but also be compliant to avoid damaging the door during the pulling
↪  process.

## Assistant
Certainly! I'm good at writing callback functions for ports in PyDrake. I will follow your instructions
↪  carefully and think carefully.

## Meta-Control
Now please implement callback functions of a `LeafSystem` that acts as a connector which I have already
↪  constructed. The input ports correspond to inputs to the skill (observations and controller
↪  parameters). The output ports correspond to the input ports of the task controller.
Here is a yaml file describing what the inputs to the skill are:
{skill_input_summary}

Here we print the port name, type and sample value for all the input port. You can get the value of the
↪  port by `value = self.GetInputPort(port_name).Eval(context)` when impelementing the call back
↪  functions:
{input_port_sample}

The chosen task model, tracking model, task controller, and tracking controller are
```python

task_model = setup_placeholder_model()

tracking_model = setup_arm_with_objects_model(arm_type, object_info)

task_controller = CartesianInterpolationController(model=task_model)

tracking_controller = CartesianStiffnessController(model=tracking_model)

```

Here is the design summary to explain the expected way of how do the controllers works, and how to connect
↪  the ports:
{design_summary} (from Design via Templates)

Here is the summary of the input ports of the task controller:
{task_controller_input_summary} (from the controller summary)

The functions you need to implement are:
```python
def task_controller_parameters(self, context, output: AbstractValue)
    pass

def task_measured_ee_pose(self, context, output: BasicVector)
    pass

def task_execution_time(self, context, output: BasicVector)
    pass

def task_target_position(self, context, output: BasicVector)
    pass

```
Here is an example response:

<task_callback>
```python
def place_holder_function(self, context, output: BasicVector):
    abstract_input_port_value = self.GetInputPort(port_name).Eval(context)
    output.SetFromVector(abstract_input_port_value["place_holder_key"])
def place_holder_function2(self, context, output: BasicVector):
    vector_input_port_value = self.GetInputPort(port_name).Eval(context)
    out = np.zeros(6)
    out[3] = vector_input_port_value[0]
    output.SetFromVector(out)
def place_holder_function3(self, context, output: AbstractValue):
    value = self.GetInputPort(port_name).Eval(context)
    output.set_value(value)
```

Please implement all the task callback functions in one code block begin with <task_callback>.

Here are some tips to implement the callback functions:
* When use a port, make sure the name of the port is correct and the port is avialble in the current
  ↪  function.
* Note that some of the ports should be passed through without any modification like control_parameters.
  ↪  Some of the ports need to be modified.
* Make sure the dimensions of the output matches the port. The output value may be only partially
  ↪  available from input port. You need to decide the value for the remainings. Be careful of which
  ↪  dimension to set value, make sure it complies with the description of the port in the controller
  ↪  summary.
* For pose, velocity and force vectors, rotation always comes first, such as [roll, yaw, pitch, x, y, z]
  ↪  or [x-rotation, y-rotation, z-rotation, x-translation, y-translation, z-translation].
* If the input port is a dictionary, make sure only use keys that has shown in the printed port value
  ↪  sample.
* If the output should be a enum type, you can use the enum type mentioned in the controller summary
  ↪  directly without any import.
* Recall the task requirements given in the beginning. Make sure the call back functions correctly solve
  ↪  the task.
* Please avoid using placeholders or simplified examples that require modifications by others. Ensure all
  ↪  calculations are conclusive. In case of any uncertainties about parameters, utilize the information
  ↪  provided to derive the most informed inference available.
* You can use `self.num_q` to denote the number of joints

Please implement the call-back functions. Please think step by step then write the code.

## Assistant
<task_callback>
```python
def task_controller_parameters(self, context, output: AbstractValue):
    # Simply pass through the controller parameters from the input to the output
    controller_params = self.GetInputPort("controller_parameters").Eval(context)
    output.set_value(controller_params)

def task_measured_ee_pose(self, context, output: BasicVector):
    # The output directly maps from the input measured EE pose, no modification needed
    ee_pose = self.GetInputPort("measured_ee_pose").Eval(context)
    output.SetFromVector(ee_pose)

def task_execution_time(self, context, output: BasicVector):
    # For this placeholder, we'll simulate a simple execution time. In practice, this
    # should be calculated based on the system time or trajectory progress.
    # Here we just output a constant value, in real applications, this needs to be dynamic.
    execution_time = [1.0]  # A placeholder value for the execution time
    output.SetFromVector(execution_time)

def task_target_position(self, context, output: BasicVector):
    # Calculate the target position based on the door's handle information
    measured_body_poses = self.GetInputPort("measured_body_poses").Eval(context)
    handle_pose = measured_body_poses['handle'].translation()

    # Given the task is to open the door, let's assume it swings about 90 degrees.
    # We know the door's handle is roughly at [0.8, 0.2] when closed.
    # For simplification, during the swing, the radius remains constant, but the door rotates.
    # Hence, we adjust the target position to mimic the door's handle at about 90 degrees open.
    # This is a simplification and should be fine-tuned based on real door mechanics.

    # Assuming an average radius for calculation simplicity.
    avg_radius = (0.3 + 0.5) / 2
    target_x = 0.8 - avg_radius  # Subtract avg_radius because we are rotating anticlockwise
    target_y = 0.2 - avg_radius  # Assume the handle stays at the same height level

    target_position = [target_x, target_y, handle_pose[2]]  # Maintaining the handle's height
    output.SetFromVector(target_position)
```

## Meta-Control
Now please implement callback functions of a `LeafSystem` that acts as a connector which I have already
  ↪  constructed for the tracking controller. The input ports correspond to inputs to the skill
  ↪  (observations and controller parameters) and task control (the output of the task controller). The
  ↪  output ports correspond to the input ports of the tracking controller.

Recall the input port of the skill given previously, and the following is a description of the task
  ↪  control port. You can use all skill input ports and the task control port.
```yaml
- description: Nominal end-effector pose based on the trajectory evaluated at the
    current time. RPY_XYZ
  name: task_ee_pose_nom
  size: 6
  type: BasicVector
```

```
```

Here is the summary of the input ports of the tracking controller:
```yaml
- description: Desired end effector target (pose or twist). [roll, pitch, yaw, x,
    y, z]
  name: ee_target
  size: 6
  type: BasicVector
- description: Type of the end effector target. This type is enum. It must be EndEffectorTarget.kPose
    or EndEffectorTarget.kTwist. Do not use `str`, use enum directly.
  name: ee_target_type
  type: AbstractValue
- description: Current joint position of the robot arm.
  name: arm_position
  size: num_q
  type: BasicVector
- description: Current joint velocity of the robot arm.
  name: arm_velocity
  size: num_q
  type: BasicVector
- description: The controller can be parameterized through an abstract input port
    "controller_parameters" to modify its behavior during runtime, allowing changes
    to cartesian_stiffness and "cartesian_damping" matrices. [roll, pitch, yaw, x,
    y, z]
  example_value:
    cartesian_stiffness: '[0.1]*3 + [200]*3'
  name: controller_parameters
  type: AbstractValue
```

The functions you need to implement are:
```python
def tracking_ee_target(self, context, output: BasicVector):
    pass

def tracking_ee_target_type(self, context, output: AbstractValue):
    pass

def tracking_arm_position(self, context, output: BasicVector):
    pass

def tracking_arm_velocity(self, context, output: BasicVector):
    pass

def tracking_controller_parameters(self, context, output: AbstractValue):
    pass
```

Here is an example response:

<tracking_callback>
```python
def place_holder_function(self, context, output: BasicVector):
    abstract_input_port_value = self.place_holder_port.Eval(context)
    output.SetFromVector(abstract_input_port_value["place_holder_key"])
def place_holder_function2(self, context, output: BasicVector):
    vector_input_port_value = self.place_holder_port.Eval(context)
    out = np.zeros(6)
    out[3] = vector_input_port_value[0]
    output.SetFromVector(out)
def place_holder_function3(self, context, output: AbstractValue):
    value = self.place_holder_port.Eval(context)
    output.set_value(value)
```

Follow the previously given tips for implementing the callback functions. Further more:
* Note that output ports of the task controller have been renamed by adding prefix: 'task_' + port_name.

Now please implement the call-back functions. Please think step by step then write the code.

<tracking_callback>
```python
def tracking_ee_target(self, context, output: BasicVector):
    # For the tracking controller, the ee_target comes directly from the task controller's output
    ee_target = self.GetInputPort("task_ee_pose_nom").Eval(context)
    output.SetFromVector(ee_target)
```

```python
def tracking_ee_target_type(self, context, output: AbstractValue):
    # As per the task, the target type is specifically for pose in this context
    # Directly setting the target type to EndEffectorTarget.kPose as required
    output.set_value(EndEffectorTarget.kPose)

def tracking_arm_position(self, context, output: BasicVector):
    # The arm position for the tracking controller comes from the skill's input port
    arm_position = self.GetInputPort("measured_arm_position").Eval(context)
    output.SetFromVector(arm_position)

def tracking_arm_velocity(self, context, output: BasicVector):
    # The arm velocity for the tracking controller also comes from the skill's input port
    arm_velocity = self.GetInputPort("measured_arm_velocity").Eval(context)
    output.SetFromVector(arm_velocity)

def tracking_controller_parameters(self, context, output: AbstractValue):
    # Passing through the controller parameters directly from the skill's input
    controller_parameters = self.GetInputPort("controller_parameters").Eval(context)
    # Assuming manipulation is needed for compliance characteristics, it could look like:
    # Modify for stiffness and damping if needed, here directly passing through for simplicity
    output.set_value(controller_parameters)
```

## Meta-Control
Here is the output of running the skill:

Human expert supplementary instructions:

Do you think it is working as expected? (Empty output means the system is running without errors)
If it is, please answer with one word "yes".
If it is not, please identify which step(s) went wrong and explain how to fix it by including which
↪  step(s) to modify (<step_name>) and the updated code (enclosed by ```python```). We have proceeded the
↪  following steps:
- task_callback
- tracking_callback

Please do not makeup an inexist step.

If the system is not working as expected, you must modify as least one step. You can modify multiple
↪  steps.
For each step, you can only modify the function that you think is wrong. Others will be kept unchanged.
Remember to add "<step_name>" before the corresponding code block. A correct solution exists with the
↪  given infomation. Try your best.

Here is a sample response when it is not working as expected:

I think this error occured because [some reason].

updated_code:
<tracking_callback>

```python
def tracking_arm_position(self, context, output: BasicVector):
    ...
```

## Assistant
yes

## Meta-Control
Please summarize the task_controller, tracking_controll, task_callback and the tracking_callback. Explain
↪  the data flow. This response will be used as prompt to choose the performance metric from
↪  measurements, and tune the controller parameters, so please include all necessary information.

## Assistant
To accomplish the task of pulling open a door using a robotic system, we've implemented a hierarchical
↪  control structure comprising a task controller and a tracking controller, along with their callback
↪  functions to connect and preprocess inputs and outputs accordingly. Here's a comprehensive summary:

### Task Controller: CartesianInterpolationController

**Purpose**: Generates a nominal end-effector pose trajectory for the robot to follow, based on the
↪  initial and target positions, ensuring the door opens approximately 90 degrees without requiring
↪  forceful interaction.

**Input Ports and Callbacks**:

- `controller_parameters`: Directly passed through from the skill inputs, it includes dynamic adjustments
  ↪ such as trajectory flags and stiffness parameters.
    - **Callback**: Passes through `controller_parameters` without modification.
- `measured_ee_pose`: Provides the current pose of the end effector in RPY_XYZ format.
    - **Callback**: Directly maps the `measured_ee_pose` from skill inputs to task controller inputs without
    ↪ modification.
- `execution_time`: An abstract concept used to evaluate the current position along the interpolated
  ↪ trajectory.
    - **Callback**: Sets a constant execution time as a placeholder; in practical applications, this should
    ↪ dynamically reflect progress along the trajectory.
- `target_position`: Determines the final target position `[X, Y, Z]` of the end effector, essential for
  ↪ calculating the trajectory that guides the door opening.
    - **Callback**: Calculates `target_position` based on the average radius estimation from the door hinge
    ↪ and adjusts for a 90-degree opening movement.

**Output Port**:
- `ee_pose_nom`: The desired end-effector pose `[roll, pitch, yaw, x, y, z]` for the robot to achieve,
  ↪ acting as the target for the tracking controller.

### Tracking Controller: CartesianStiffnessController

**Purpose**: Executes the movement towards the target pose with specified compliance to prevent damage,
↪ utilizing joint torques calculated based on the current and desired poses.

**Input Ports and Callbacks**:
- `ee_target`: The desired end-effector pose for the robot to track.
    - **Callback**: Receives `ee_pose_nom` from the task controller and directly passes it.
- `ee_target_type`: Specifies that the target is a pose (instead of a twist), dictating the controller's
  ↪ focus.
    - **Callback**: Sets the target type to `EndEffectorTarget.kPose`.
- `arm_position` and `arm_velocity`: Provide current joint positions and velocities of the robot's arm.
    - **Callback**: Direct mappings from skill inputs to tracking controller inputs without modifications.
- `controller_parameters`: Allows runtime modifications to control behavior, particularly to stiffness and
  ↪ damping matrices.
    - **Callback**: Passes `controller_parameters` from skill inputs to tracking controller inputs directly.

### Data Flow:

1. **Skill Inputs**: Incoming observations and parameters (like measured joint states, arm
   ↪ positions/velocities, and controller configurations) feed into both the task and tracking controllers.
2. **Task Callbacks Process**: These functions preprocess and appropriately route data from skill inputs
   ↪ to the task controller. They are responsible for setting up the nominal trajectory by determining
   ↪ execution time and target positions.
3. **Task Controller**: With its inputs configured, calculates the desired end-effector pose.
4. **Tracking Callbacks Process**: Receives the nominal pose from the task controller along with other
   ↪ necessary parameters and skill inputs. Prepares data for the tracking controller.
5. **Tracking Controller**: Utilizes the preprocessed inputs to compute the appropriate control actions,
   ↪ aiming to track the specified end-effector pose trajectory compliantly.

This entire process ensures the robot can adjust its approach dynamically to open the door by
↪ approximately 90 degrees, based on the initial observations and ongoing measurements, all while being
↪ compliant to avoid damage. The performance metric, to be chosen from measurements, could focus on the
↪ accuracy of the end-effector pose in reaching the target position, compliance in execution to avoid
↪ excessive force, and efficiency in trajectory execution time. Tuning controller parameters,
↪ particularly within `controller_parameters` for both task and tracking stages, will be crucial in
↪ optimizing these aspects.

## C.4   Model template summary

```
setup_linear_model:
  description: |
    This function returns an affine dynamic model.
    This model is especially useful when the system is or can be approximated by affine dynamics.
    The dynamic equation is: x_dot = A * x + B * u;  y = C * x + D * u.

    Example usage:
    """
    x0 = np.zeros((4,1))
    u0 = np.zeros((1,1))
    m = 0.01
    l = 0.1
    g = 0.81
    A = np.array([[0,1,0,0], [0,0,m*g,0],[0,0,0,1],[0,0,0,1]])
    B = np.array([[0],[1/m],[0],[-1/m]])
    C = np.eye(4)
    D = np.zeros((4, 1))
```

```yaml
        model = setup_linear_model(x0, u0, A, B, C, D)
        """
    args:
      - name: x0
        type: numpy.ndarray
        size: (n_x, 1)
        description: Initial state
      - name: u0
        type: numpy.ndarray
        size: (n_u, 1)
        description: Initial input
      - name: A
        type: numpy.ndarray
        size: (n_x, n_x)
        description: State matrix
      - name: B
        type: numpy.ndarray
        size: (n_x, n_u)
        description: Input matrix
      - name: C
        type: numpy.ndarray
        size: (n_y, n_x)
        description: Output matrix
      - name: D
        type: numpy.ndarray
        size: (n_y, n_u)
        description: Feedforward matrix

setup_arm_model:
  description: |
    This function returns a model of the robot arm.
    This model is especially useful when the controller only needs to consider the robot arm.
    Because this function will be called in skill's init function, arm type is directly available
    as arm_type.
    Usage:
    """
    model = setup_arm_model(arm_type)
    """
  args:
    - name: arm_type
      type: str
      description: The robot arm type. It should be consistent as the skill's.

setup_arm_with_objects_model:
  description: |
    This function returns a model of the robot arm and static objects in the scene.
    This model is especially useful when the controller needs to consider the interaction between
    the robot and objects, such as grasping and collision avoidance.
    It can be used for controlling in the cartesian space or in the joint space.
    Because this function will be called in skill's init function, arm type and object_info
    are directly available as arm_type and object_info.
    Usage:
    """
    model = setup_arm_with_objects_model(arm_type, object_info)
    """
  args:
    - name: arm_type
      type: str
      description: The robot arm type. It should be consistent as the skill's.
    - name: object_info
      type: dict
      description: The dictionary of all objects in the scene.  It should be consistent as the skill's.

setup_placeholder_model:
  description: |
    This function returns None.
    This model is especially useful when the controller does not need any dynamic model.
    Example usage:
    """
    model = setup_placeholder_model()
    """
```

## C.5   Controller template summary

```yaml
LQRController:
  description: |
    This class implements a Linear Quadratic Regulator (LQR) controller for a given affine system.
```

```
    The controller computes control inputs based on the state deviations from a given equilibrium.
  args:
    - name: model
      type: Diagram
      description: A Diagram containing the AffineSystem for which the LQR controller is designed.
    - name: Q
      type: numpy.ndarray
      description: State cost matrix.
    - name: R
      type: numpy.ndarray
      description: Control cost matrix.
    - name: x0
      type: numpy.ndarray
      description: Equilibrium state around which the controller is designed.
    - name: u0
      type: numpy.ndarray
      description: Equilibrium control input.
  input_ports:
    - name: state
      type: BasicVector
      size: "len(x0)"
      description: Represents the current state of the system.
  output_ports:
    - name: control
      type: BasicVector
      size: "len(u0)"
      description: Represents the computed control input based on the LQR control law.

IdentityController:
  description: |
    This class implements an identity controller that directly outputs the arm torque it receives.
    It is the best choice when the task controller is already enough to accomplish the task and we
    only need a placeholder for the tracking controller.
  args:
    - name: model
      type: Diagram
      description: A Diagram containing the MultiBodyPlant model of the robotic system.
  input_ports:
    - name: applied_arm_torque
      type: BasicVector
      size: "plant.num_actuators()"
      description: applied_arm_torque computed by the task controller.
  output_ports:
    - name: applied_arm_torque
      type: BasicVector
      size: "plant.num_actuators()"
      description: directly output the input applied_arm_torque.

PoseForceController:
  description: |
    This class implements a controller that combines pose and force control for a robotic arm.
    The controller computes torques based on the desired end-effector pose and force.
    It is best suitable for the situation when we have a desired force on some dimensions and a desired
↪   position or rotation on other dimensions. Note that the controller can not track force and pose at the
↪   same time for a dimension.
  args:
    - name: model
      type: Diagram
      description: A Diagram containing the MultiBodyPlant model of the robotic system.
    - name: force_control_axis_mask
      type: numpy.ndarray
      description: Boolean array of length 6 to specify which axes are controlled by force. The order is
↪   [x-rotation, y-rotation, z-rotation, x-translation, y-translation, z-translation]. For each dimension,
↪   True represents force control, False represents pose control.
  input_ports:
    - name: ee_pose_target
      type: BasicVector
      size: 6
      description: Desired end-effector pose (roll, pitch, yaw, x, y, z).
    - name: ee_force_target
      type: BasicVector
      size: 6
      description: Desired end-effector force. (x-rotation, y-rotation, z-rotation, x-translation,
↪   y-translation, z-translation).
    - name: arm_position
      type: BasicVector
      size: "plant.num_positions(arm)"
      description: Current joint positions of the arm.
    - name: arm_velocity
      type: BasicVector
```

```yaml
      size: "plant.num_velocities(arm)"
      description: Current joint velocities of the arm.
  output_ports:
    - name: applied_arm_torque
      type: BasicVector
      size: "plant.num_actuators()"
      description: Computed joint torques for the robot arm.

CartesianStiffnessController:
  description: A cartesian stiffness controller using impedance control to determine control torques.
↪  Makes the robot behave as if a spring-damper system is attached to the end-effector. The stiffness can
↪  be adjusted dynamically by providing input to the controller_parameters port. The advantage of this
↪  controller is it can provide compliant behavior with adjustable stiffness.
  args:
    - name: model
      type: Diagram
      description: A Diagram containing the robot model (MultiBodyPlant) for computing dynamics
  input_ports:
    - name: ee_target
      type: BasicVector
      size: 6
      description: Desired end effector target (pose or twist). [roll, pitch, yaw, x, y, z]
    - name: ee_target_type
      type: AbstractValue
      description: Type of the end effector target. This type is enum. The must be EndEffectorTarget.kPose
↪  or EndEffectorTarget.kTwist. Do not use `str`, use enum directly.
    - name: arm_position
      type: BasicVector
      size: "(number of positions of the plant's arm)"
      description: Current joint position of the robot arm.
    - name: arm_velocity
      type: BasicVector
      size: "(number of velocities of the plant's arm)"
      description: Current joint velocity of the robot arm.
    - name: controller_parameters
      type: AbstractValue
      example_value:
        cartesian_stiffness: "[0.1]*3 + [200]*3"
      description: The controller can be parameterized through an abstract input port
↪  "controller_parameters" to modify its behavior during runtime, allowing changes to cartesian_stiffness
↪  and "cartesian_damping" matrices. [roll, pitch, yaw, x, y, z]
  output_ports:
    - name: applied_arm_torque
      type: BasicVector
      size: "(number of actuators in the plant)"
      description: Control torques applied to the joints of the robot arm.

SafeController:
  description: A controller that projects nominal control torque to a safe control torque using a control
↪  barrier function. Currently, the safe constraint is collision avoidance. The advantage of this
↪  controller is to safeguard a nominal control input port to realize reactive collision avoidance.
  args:
    - name: model
      type: Diagram
      description: Diagram containing the multibody plant and scene graph, as well as the information of
↪  obstacles for collision avoidance.
    - name: meshcat
      type: Meshcat
      description: This arg is optional. For visualization purposes.
  input_ports:
    - name: arm_target
      type: BasicVector
      size: "num_q"
      description: Desired joint target for the arm.
    - name: arm_target_type
      type: AbstractValue
      description: Type of the target. This type is enum. The value must be JointTarget.kPosition,
↪  JointTarget.kVelocity, or JointTarget.kTorque. Do not use `str`, use enum directly.
    - name: arm_position
      type: BasicVector
      size: "num_q"
      description: Current joint position of the arm.
    - name: arm_velocity
      type: BasicVector
      size: "num_q"
      description: Current joint velocity of the arm.
  output_ports:
    - name: applied_arm_torque
      type: BasicVector
      size: "num_q"
      description: Computed joint torque for the arm.
```

```yaml
CartesianTrajectoryController:
  description: Constructs a trajectory in cartesian space and publishes waypoints by interpolating the
↪  trajectory. Depending on the polynomial order, it can produce first-order holds or cubic splines for
↪  the trajectory.
  args:
    - name: model
      type: Diagram
      description: model is not useful in this controller, but should be passed in for consistency
    - name: polynomial_order
      default: 1
      description: Order of the polynomial for trajectory optimization. Supported values are 1 (for First
↪  Order Hold) and 3 (for Cubic Spline).
    - name: meshcat
      type: Meshcat
      description: A visualization tool. (optional)
  input_ports:
    - name: controller_parameters
      type: AbstractValue
      example_value:
        new_trajectory: true
        way_points:
          times: "[list of times]"
          points: "[list of pose in RPY_XYZ format]"
      description: Controller parameters that can indicate the need for a new trajectory and provide
↪  waypoints.
    - name: measured_ee_pose
      type: BasicVector
      size: 6
      description: Measured pose of the end effector in RPY_XYZ format.
    - name: measured_ee_twist
      type: BasicVector
      size: 6
      description: Measured twist of the end effector. RPY_XYZ
  output_ports:
    - name: ee_pose_nom
      type: BasicVector
      size: 6
      description: Nominal end-effector pose based on the last solved trajectory spline evaluated at the
↪  current time. RPY_XYZ

KinematicTrajectoryModelPredictiveController:
  description: An MPC controller using kinematic trajectory optimization to calculate a collision-free
↪  trajectory in joint space. The objective of this controller is to reach a goal in cartesian space with
↪  the end effector. Please note that the output is in joint space.
  args:
    - name: model
      type: Diagram
      description: Diagram containing the multibody plant and scene graph, as well as the information of
↪  obstacles for collision avoidance
    - name: resolve_period
      type: float
      default: 3.0
      description: Trajectory optimization resolution period.
    - name: num_steps
      type: int
      default: 20
      description: Number of optimization steps.
    - name: meshcat
      type: Meshcat Optional
      description: For visualization
  input_ports:
    - name: controller_parameters
      type: AbstractValue
      example_value:
        goal_pose: RigidTransform
      description: Controller parameters including desired goal pose for the end effector.
    - name: measured_arm_position
      type: BasicVector
      size: "(number of positions of the robot arm)"
      description: Current joint position of the robot arm.
    - name: measured_arm_velocity
      type: BasicVector
      size: "(number of velocities of the robot arm)"
      description: Current joint velocity of the robot arm.
  output_ports:
    - name: q_nom
      type: BasicVector
```

```
        size: "(number of actuators in the plant)"
        description: Nominal joint positions of the arm.
```

## C.6 Skill input summary

```
- port_name: controller_parameters
  description: Controller parameters that can be adjusted dynamically during running.
  type: AbstractValue
  data_structure: dict

- port_name: measured_arm_position
  description: Observation of the robot arm's joint position.
  type: BasicVector
  dimensions: num_q

- port_name: measured_arm_velocity
  description: Observation of the robot arm's joint velocity.
  type: BasicVector
  dimensions: num_q

- port_name: measured_ee_pose
  description: Observation of the end effector's pose. The first three elements correspond to rotation.
↪   The last three elements correspond to translation.
  type: BasicVector
  dimensions: 6

- port_name: measured_ee_twist
  description: Observation of the end effector's twist. The first three elements correspond to rotation.
↪   The last three elements correspond to translation.
  type: BasicVector
  dimensions: 6

- port_name: measured_joint_states
  description: Observation of the joint states (including joint states of articulated objects).
  type: AbstractValue
  data_structure: dict

- port_name: measured_body_poses
  description: Observation of the poses of the rigid bodies in the scene.
  type: AbstractValue
  data_structure: dict

- port_name: measured_body_velocities
  description: Observation of the velocities of the rigid bodies in the scene.
  type: AbstractValue
  data_structure: dict
```

## C.7 Input port sample

```
=========================================
TrackingControllerConverter input port contents:
controller_parameters : <class 'dict'> , {'gripper_command': 'release'}
measured_arm_position : <class 'numpy.ndarray'> , [ 0.         0.26179939  3.14159265 -2.26918531  0.
↪   0.9599
  1.57079633]
measured_arm_velocity : <class 'numpy.ndarray'> , [0. 0. 0. 0. 0. 0. 0.]
measured_ee_pose : <class 'numpy.ndarray'> , [ 1.57108469e+00 -3.92246873e-06  1.57079455e+00
↪   5.96598901e-01
  1.34625984e-03  4.33589236e-01]
measured_ee_twist : <class 'numpy.ndarray'> , [0. 0. 0. 0. 0. 0.]
measured_joint_states : <class 'dict'> , {'Actuator1': {'positions': array([0.]), 'velocities':
↪   array([0.])}, 'Actuator2': {'positions': array([0.26179939]), 'velocities': array([0.])}, 'Actuator3':
↪   {'positions': array([3.14159265]), 'velocities': array([0.])}, 'Actuator4': {'positions':
↪   array([-2.26918531]), 'velocities': array([0.])}, 'Actuator5': {'positions': array([0.]),
↪   'velocities': array([0.])}, 'Actuator6': {'positions': array([0.9599]), 'velocities': array([0.])},
↪   'Actuator7': {'positions': array([1.57079633]), 'velocities': array([0.])}, 'EndEffector':
↪   {'positions': array([], dtype=float64), 'velocities': array([], dtype=float64)},
↪   'world_welds_to_base_link': {'positions': array([], dtype=float64), 'velocities': array([],
↪   dtype=float64)}, 'PolePin': {'positions': array([0.1]), 'velocities': array([0.])},
↪   'end_effector_link_welds_to_Cart': {'positions': array([], dtype=float64), 'velocities': array([],
↪   dtype=float64)}, '$world_obstacle_box_2': {'positions': array([ 0.99875026,  0.        ,  0.        ,
↪   -0.04997917,  0.4        ,
```

```
      0.3      ,  0.101    ]), 'velocities': array([0., 0., 0., 0., 0., 0.])}, '$world_goal_box':
↪ {'positions': array([1.   , 0.   , 0.   , 0.   , 0.6  , 0.   , 0.026]), 'velocities': array([0., 0.,
↪ 0., 0., 0., 0.])}}
measured_body_poses :  <class 'dict'> ,  {'Cart': RigidTransform(
  R=RotationMatrix([
    [1.771814082172029e-06, 0.9999999584203277, -0.0002883681730920518],
    [0.9999999999907371, -1.7706828933479899e-06, 3.922979501902304e-06],
    [3.9224687301950226e-06, -0.00028836818003996167, -0.9999999584142024],
  ]),
  p=[0.48659890557829427, 0.001346454619425365, 0.4336209565978073],
)}
measured_body_velocities :  <class 'dict'> ,  {'Cart': SpatialVelocity(
  w=[0.0, 0.0, 0.0],
  v=[0.0, 0.0, 0.0],
)}

task_control :  <class 'numpy.ndarray'> ,  [0.]
===============================================
```

