# OpenReview forum: "Meta-Control: Automatic Model-based Control Synthesis for Heterogeneous Robot Skills"
_robot-learning.org/CoRL/2024/Conference — CoRL 2024_

### Official Review · Reviewer_U2Js · 2024-07-19
**Meta-Control: Automatic Model-based Control Synthesis for Heterogeneous Robot Skills**

**Originality:** 3
**Technical Quality:** 3
**Clarity Of Presentation:** 2
**Potential Impact:** 3
**Recommendation:** 3
**Confidence:** 2

**Review:**

The paper is interesting and from the start provides a rather extensive review on approaches utilizing LLMs to control robots in Section 2.
Then, in Section 3, the pipeline that involves three levels is introduced: the strategy level, the data flow level, and the parameter level. The three-level design effectively extracts control knowledge from LLM to facilitate synthesis in a Socrates’ “art of midwifery” manner. This Socrates’ “art of midwifery manner” is mentioned several times, but not explained properly.
Overall, while the paper shows impressive results in automatically obtaining code for control and execution of the tasks, it is written in a manner that is not fully clear. To the best of my knowledge, it seems that providing a proper query input into the chatGPT and some carefully chosen priors, it can generate code that will successfully execute tasks in a predefined environment/control framework. The work is evaluated in several examples, even real world, but from the paper itself it is somewhat unclear, as a lot of the text comes down to fawning over how good the process is, but not really on how to apply it. More is clear from the extensive appendices.
Overall, it is interesting and impressive, but the level of presentation is somewhat skewed towards saying how good all is.

**Quality Of The Limitations Section:**

2

**Questions For Rebuttal:**

see above

**Robotics Focus:**

4

**Summary Of Paper:**

The paper presents LLM-enabled automatic control synthesis approach that  creates customized state representations and control strategies tailored to specific tasks. The pipeline is implemented with PyDrake.

**Summary Of Recommendation:**

Overall, it is interesting and impressive, but the level of presentation is somewhat skewed towards saying how good all is.

---

### Official Review · Reviewer_5Tm1 · 2024-07-22
**A great evolution from previous works on LLM for policies (e.g. code-as-policies, language-to-rewards)**

**Originality:** 5
**Technical Quality:** 2
**Clarity Of Presentation:** 2
**Potential Impact:** 4
**Recommendation:** 3
**Confidence:** 4

**Review:**

QUALITY
The paper has an overall good quality. The main body is relatively high level and most of the interesting technical details are in the appendix (especially in the transcript of the chat with GPT).
CLARITY
The per is clear and well written but the main body of the paper is high-level with no technical details. The main technical contribution (e.g. LLM-prompting, templates, problem specification) and the resulting answers (e.g. the LLM output) are in the appendix which is however not as clear as it should be. The achieved results aren't therefore very clear and the reader lacks the detail for replicating the results. Authors have copy-pasted a conversation with the LLM for solving the "stabilize a cart pole" problem but this is just a specific example rather than a general procedure (even though part of the conversation seem to contain some "boiler plate" promting)
ORIGINALITY
Even if the paper is a follow up from previous papers (e.g. code-as-policies, language-to-rewards), it has a clear originality, especially in the explicit connection with classical control theory.
SIGNIFICANCE
A clear step forward with respect to previous papers. This paper has the potential to have an impact on research.
STRENGTHS
The paper uses in a very clever way the semantic understanding of LLMs together with the benefits of classical control tools like linear systems, linear quadratic regulators, etc.
WEAKNESSES
It's unclear what the exact steps of the proposed approach are. The main body of the paper lacks the technical details. The appendix contains the conversation with the language model but this isn't enough to retrieve the exact steps to the followed in order to replicate all the results of the paper.

**Quality Of The Limitations Section:**

2

**Questions For Rebuttal:**

As a reviewer, I couldn't really understand how to replicate the proposed results. The body of the paper doesn't give enough details so I have tried to retrieve these details from the appendix. However, the appendix is unclear. The conversation seems to have 3 different users (named: "## Assistant", "## User" and "## Meta-Control"). Also, the authors do not give a full description of how MetaControl works (i.e. produces its own text output): some messages seem to be just a stereotyped prompts "The GPT is a professor in robot control and a proficient programmer..."; some other messages seem to be a copy-paste of carefully chosen previous assistant answers "The chosen task model, tracking model, task controller, and tracking controller are..."). In order for others to replicate the given results, authors should give a complete description of how Meta-Control works, possibly describing it in the form of finite state machine (with parameters), so that it can be coded in some programming language.

**Robotics Focus:**

4

**Summary Of Paper:**

The paper proposes an approach to use LLM for controller synthesis, using classical concepts like state-space linear systems, LQ regulators, MPC etc

**Summary Of Recommendation:**

Very interesting paper but, in its current version, very difficult to replicate systematically

---

### Official Review · Reviewer_9Tq5 · 2024-07-27
**Review of Submission 415**

**Originality:** 3
**Technical Quality:** 3
**Clarity Of Presentation:** 4
**Potential Impact:** 3
**Recommendation:** 3
**Confidence:** 2

**Review:**

**Strengths:**
- The main contribution of the paper is the evaluation of LLMs as designers of model-based controllers, which is, to the best of my knowledge, a novel and interesting avenue, with clear practical applications.
- The presentation is overall polished and clear, with minor issues as detailed in the questions below.

**Weaknesses:**
- One significant issue is that the author's process for prompt design is not clear. In my opinion, this work would largely benefit if a prompt designed for a single task (say, the balancing one) could be easily reused for other tasks by simply replacing the task description (e.g. the first user query in Appendix C.1). It is not clear whether the proposed prompt was instead designed while considering feedback from all three tasks. If the prompt is not easily adaptable to an unseen task, then the method would mostly redirect human supervision from controller design to prompt design, and be thus much less impactful. In other words, without empirical evidence for generalization to new tasks without tuning the prompt, this paper would mostly show that, given a fixed task, it is possible to design a prompt that guides a LLM to synthesize a working controller. I remain open to update my score if the authors address this point.
- A further issue regards the breadth of the experiments. Due to the modularity and flexibility of the framework, it should be possible to evaluate the method across a much greater number of tasks, models and controllers. I would suggest the authors to consider additional manipulation tasks, for instance.
- The discussion on limitations mostly focuses on the issues of model-based controllers, rather than on the issues of LLM-guided controller synthesis. The paper would thus benefit from an extended discussion of failure cases of the LLM (on top of those presented in section 4.4). For instance, did the authors provide other controllers which the LLM failed to use?

On top of these main points, there are several others in the questions below.

**Post-rebuttal:**
The authors presented additional evidence to support generalization, in the form of evaluations on two additional tasks. They also updated the discussion on limitation, which improves the paper's placement. I now lean towards acceptance. I would recommend authors to add additional information to the manuscript, as discussed during the rebuttal.

**Quality Of The Limitations Section:**

2

**Questions For Rebuttal:**

**Questions:**
- How many shots are necessary for few-shot finetuning in each task? This information is currently missing and would be very helpful.
- How would the author scale as more controllers/models are provided, and choosing the right controller/model becomes harder?
- The author mention slow synthesis as a limitation. Could they also provide the runtime for the method?
- What do the authors mean by "art of midwifery"? This term has a rather clear method in philosophy, but it is not entirely clear how MetaControl embodies this principle.
- Line 125: why is $g$ considered a dynamic model? To the best of my knowledge, it does not represent a state change.

**Robotics Focus:**

4

**Summary Of Paper:**

This work presents an LLM-guided synthesis process for hierarchical model-based controllers. In a first query, the LLM is queried with a task provided in natural language and asked to select a high-level task model and task controller, as well as a low-level tracking model and tracking controller from a library provided in-context. This first step also provides parameters for controllers and models. The second query asks the LLM to synthesize interfaces between perception and the two controllers. These two queries are then followed by an optimization loop in which the proposed controller is executed, and the LLM is presented with the resulting trajectory, and asked to update the paramters it previously selected. The combination of the three steps results in a method, named MetaControl. The resulting controller is able to solve three tasks in simulation (an arm-bound pendulum, a door opening and a pick-place task), as well as a board erasing task on real hardware. Each of the three components is ablated, highlighting in particular the necessity of few-shot prompting to improve the initial parameters suggested by the LLM. As the final product of MetaControl is an established model-based controller, the method benefits from interpretability and generalization (e.g. by replacing the robot's model).

**Summary Of Recommendation:**

Despite the good presentation and significance of this work, there is currently lack of clarity on how the prompts were designed (on a single task, or on multiple) and on how the method would be able to generalize in a zero-shot fashion. Thus, I would lean towards rejection, but I remain open to updating my evaluation.

---

### Author Rebuttal · Authors · 2024-08-10

Dear Area Chair and Reviewers,

Thank you for your thorough and constructive feedback! We have carefully addressed all the points raised and made the following key revisions:

- **Prompt Design, Generalization, and Application**: We revised Section 3.2 and updated Figure 3 to clarify our prompt design process and how it applies to unseen tasks in a zero-shot manner without modification. Additionally, we provided more details on Meta-Control’s workflow, including an explanation of the “art of midwifery” and conversations from other tasks to enhance understanding. Figure 3 has been redrawn in a finite state machine-like manner to improve replicability and clarity. Please find the revised paper in the zip file, with changes highlighted.

- **Expanded Experiments**: To address concerns about the breadth of experiments, we added two new tasks: push-button and open-drawer. We hope these additional experiments further demonstrate the generalizability of our method. Videos of these experiments are available on the project website (meta-control-anonymous.github.io/).

- **Enhanced Discussion on Limitations**: We expanded the discussion on failure cases in the limitations section, focusing on logical and mathematical errors made by the LLM and instances where suboptimal controller choices were made.

We believe these revisions improve the clarity, replicability, and persuasiveness of our work. Thank you again for your valuable feedback, which has greatly strengthened our submission.

---

### Decision · Program_Chairs · 2024-09-04

**Decision:**

Accept

**Comment:**

The paper was initially evaluated with mixed reviews, but after the rebuttal, we the reviewers suggest accepting the paper, although without strong backing.

Here is a high level overview of the reviews. (before rebuttal)
### Strengths:
- Novel approach using LLMs for model-based controllers with clear practical applications. (Reviewer 9Tq5, 5Tm1)
- Overall polished and clear presentation (Reviewer 9Tq5), although critique of lacking details.
- Review of LLM-based control approaches. (Reviewer U2Js)
- Impressive results. (Reviewer U2Js)

### Weaknesses:
- Lack of clarity in the prompt design process and generalization (Reviewer 9Tq5) and lack of essential technical details for replication (Reviewer 5Tm1)
- Insufficient evaluation across diverse tasks and models. (Reviewer 9Tq5)
- Unclear application details and presentation. (Reviewer U2Js)

In the rebuttal, additional evidence was added and the presentation was improved. Still, reviewer 5Tm1 things the explanation of how the automated dialog could be improved.
Given the potential of the method and the overall good presentation, I suggest accepting the paper.